# THE EFFECTS OF INVERTIBILITY ON THE REPRESENTATIONAL COMPLEXITY OF ENCODERS IN VAES

**Divyansh Pareek**
Machine Learning Department
Carnegie Mellon University
Pittsburgh, PA, 15213
dpareek@andrew.cmu.edu

**Andrej Risteski**
Machine Learning Department
Carnegie Mellon University
Pittsburgh, PA, 15213
aristesk@andrew.cmu.edu

## ABSTRACT

Training and using modern neural-network based latent-variable generative models (like Variational Autoencoders) often require simultaneously training a generative direction along with an inferential (encoding) direction, which approximates the posterior distribution over the latent variables. Thus, the question arises: how complex does the inferential model need to be, in order to be able to accurately model the posterior distribution of a given generative model? In this paper, we identify an important property of the generative map impacting the required size of the encoder. We show that if the generative map is "strongly invertible" (in a sense we suitably formalize), the inferential model need not be much more complex. Conversely, we prove that there exist non-invertible generative maps, for which the encoding direction needs to be exponentially larger (under standard assumptions in computational complexity). Importantly, we do not require the generative model to be layerwise invertible, which a lot of the related literature assumes and is not satisfied by many architectures used in practice (e.g. convolution and pooling based networks). As a non-invertible generator results in a distribution supported on a low-dimensional manifold, this provides theoretical support for the empirical wisdom that learning deep generative models is harder when data lies on a sufficiently complex low-dimensional manifold.

## 1 INTRODUCTION

Many modern generative models of choice (e.g. Generative Adversarial Networks (Goodfellow et al., 2014), Variational Autoencoders (Kingma & Welling, 2013)) are modeled as non-linear, possibly stochastic transformations of a simple latent distribution (e.g. a standard Gaussian). A particularly common task is modeling the *inferential (encoder) direction*: that is, modeling the posterior distribution on the latents $z$ given an observable sample $x$. Such a task is useful both at *train time* and at *test time*. At train time, fitting generative models like variational autoencoders via maximum likelihood often relies on variational methods, which require the joint training of a *generative model (i.e. generator/decoder)*, as well as an *inference model (i.e. encoder)* which models the posterior distribution of the latent given the observables. At test time, the posterior distribution very often has some practical use, e.g. useful, potentially interpretable feature embeddings for data (Salimans et al., 2016; Berthelot et al., 2018), "intervening" on the latent space to change the sample in some targeted manner (Shen et al., 2020), etc. As such, the question of the "complexity" of the inference model (i.e. number of parameters to represent it using a neural network-based encoder) as a function of the "complexity" of the forward model is of paramount importance, so that training is computationally tractable:

**Question:** *How complex does the inference (encoder) model need to be relative to the complexity of the generative (decoder) model ?*

In this paper we identify an important property of the generative direction governing the complexity of the inference direction for *variational autoencoders*: *bijectivity/invertibility* of the mean of the generative direction. We prove that when the mean of the generative direction is invertible, the complexity of the inference direction is not much greater than the complexity of the generative

direction. Conversely, when the mean of the generative direction is not invertible, modulo standard computational complexity conjectures from cryptography, we can exhibit instances where the inference direction has to be much more complex.

On the mathematical level, our techniques involve a neural simulation of a Langevin random walk to sample from the posterior of the latent variables and uncover novel connections between Langevin diffusions and (hierarchical) deep latent Gaussians. On the lower bound side, we provide a reduction from the existence of *one-way Boolean permutations* in computational complexity: that is, permutations that are easy to calculate, but hard to invert. We show that the existence of a small encoder for non-invertible generators would allow us to design an invertor for *any* Boolean permutation, thus violating the existence a one-way permutation. This is the first time such ideas have been applied to generative models.

Note that a non-invertible generator results in a distribution supported on a lower dimensional manifold. Thus, our results show that learning deep generative models is harder when data lies on a sufficiently complex low-dimensional manifold, corroborating similarly flavored empirical observations (Dai & Wipf, 2019; Arjovsky et al., 2017).

## 2 OUR RESULTS

The Variational Autoencoder (VAE) (Kingma & Welling, 2013) is one of the most commonly used paradigms in generative models. It's trained by fitting a *generator* which maps latent variables $z$ to observables $x$, denoted by $p_\theta(x|z)$, as well as an *encoder* which maps the observables to the latent space, denoted by $q_\phi(z|x)$. Here $\phi$ and $\theta$ are the encoder parameters and generator parameters respectively. Given $n$ training samples $\{x^{(i)}\}_{i=1}^n$, the VAE objective is given by

$$\max_{\phi,\theta} \frac{1}{n} \sum_{i=1}^n \mathbb{E}_{z \sim q_\phi(.|x^{(i)})} \left[ \log p_\theta(x^{(i)}|z) \right] - KL\left( q_\phi(z|x^{(i)}) || p(z) \right)$$

where $p(z)$ is typically chosen to be a standard Gaussian. This loss can be viewed as a variational relaxation of the maximum likelihood objective, where the encoder $q_\phi$, in the limit of infinite representational power, is intended to model the posterior distribution $p_\theta(z|x^{(i)})$.

**Setup:** We will consider a setting in which the data distribution itself is given by some ground-truth generator $G : \mathbb{R}^{d_l} \to \mathbb{R}^{d_o}$, and ask how complex (in terms of number of parameters) the encoder needs to be (as a function of the number of parameters of $G$), s.t. it approximates the posterior distribution $p(z|x)$ of the generator.

We will consider two standard probabilistic models for the generator/encoder respectively.

**Definition 1** (Latent Gaussian). A *latent Gaussian* is the conditional distribution given by a stochastic pushforward of a Gaussian distribution. That is, for latent variable $z \in \mathbb{R}^{d_l}$ and observable $x \in \mathbb{R}^{d_o}$, for a neural network $G : \mathbb{R}^{d_l} \to \mathbb{R}^{d_o}$ and noise parameter $\beta^2$, we have $p(x|z) = \mathcal{N}(G(z), \beta^2 I_{d_o})$ and $p(z) = \mathcal{N}(0, I_{d_l})$.

In other words, a sample from this distribution can be generated by sampling independently $z \sim \mathcal{N}(0, I_{d_l})$ and $\xi \sim \mathcal{N}(0, \beta^2 I_{d_o})$ and outputting $x = G(z) + \xi$. This is a standard neural parametrization of a generator with (scaled) identity covariance matrix, a fairly common choice in practical implementations of VAEs (Kingma & Welling, 2013; Dai & Wipf, 2019).

We will also define a probabilistic model which is a *composition* of latent Gaussians (i.e. consists of multiple *stochastic* layers), which is also common, particularly when modeling encoders in VAEs, as they can model potentially non-Gaussian posteriors (Burda et al., 2015; Rezende et al., 2014):

**Definition 2** (Deep Latent Gaussian). A *deep latent Gaussian* is the conditional distribution given by a sequence of stochastic pushforwards of a Gaussian distribution. That is, for observable $z_0 \in \mathbb{R}^{d_0}$ and latent variables $\{z_i \in \mathbb{R}^{d_i}\}_{i=1}^L$, for neural networks $\{G_i : \mathbb{R}^{d_{i-1}} \to \mathbb{R}^{d_i}\}_{i=1}^L$ and noise parameters $\{\beta_i^2\}_{i=1}^L$, the conditional distribution $p(z_L|z_0)$ is a deep latent Gaussian when $p(z_i|z_{i-1}) = \mathcal{N}(G_i(z_{i-1}), \beta_i^2 I_{d_i}), \forall i \in [L]$ and $p(z_0) = \mathcal{N}(0, I_{d_0})$.

In other words, a deep latent Gaussian is a distribution, which can be sampled by ancestral sampling, one layer at a time. Note that this class of distributions is convenient as a choice for an encoder

in a VAE, since compositions are amenable to the reparametrization trick of Kingma & Welling (2013)—the randomness for each of the layers can be "presampled" and appropriately transformed (Burda et al., 2015; Rezende et al., 2014). Then, we ask the following:

**Question:** If a VAE generator is modeled as a latent Gaussian (that is, $p(x|z) \equiv \mathcal{N}(G(z), \beta^2 I)$), s.t. the corresponding $G$ has at most $N$ parameters, and we wish to approximate the posterior $p(z|x)$ by a deep latent Gaussian s.t. the total size of the networks in it have at most $N'$ parameters, how large must $N'$ be as function of $N$?

We will work in the setting $d_l = d_o = d$, and prove a dichotomy based on the invertibility of $G$: namely, if $G : \mathbb{R}^d \to \mathbb{R}^d$ is bijective, and $\beta \leq \mathcal{O}\left(\frac{1}{d^{1.5}\sqrt{\log d/\epsilon}}\right)$, the posterior $p(z|x)$ can be $\epsilon$-approximated in total variation distance by a deep latent Gaussian of size $N' = \mathcal{O}\left(N \cdot poly(d, 1/\beta, 1/\epsilon)\right)$. Thus, if the neural network $G$ is invertible, and for a fixed $\epsilon$ and a small-enough variance term $\beta^2$, we can approximate the posterior with a deep latent Gaussian polynomially larger than $G$. On the other hand, if $G$ is not bijective, if one-way-functions exist (a widely believed computational complexity conjecture), we will show there exists a VAE generator $G : \mathbb{R}^d \to \mathbb{R}^d$ of size polynomial in $d$, for which the posterior $p(z|x)$ cannot be approximated in total variation distance for *even an inverse polynomial fraction* of inputs $x$, unless the inferential network is of size *exponential* in $d$.

**Remark 1:** Note that it is in fact *sufficient* to investigate only the case where $d_l = d_o = d$. Let us elaborate why:

- For Theorem 1, a function $G : \mathbb{R}^{d_l} \to \mathbb{R}^{d_o}$ can be bijective, Lipschitz and strongly invertible (i.e. satisfies Assumptions 1 and 2) *only if* $d_l = d_o$, so Theorem 1 (upper bound) would not make any sense unless the dimensions are equal. (Note: there *are* bijective maps between, say, $\mathbb{R}$ and $\mathbb{R}^2$, e.g. space-filling curves, but these cannot be Lipschitz, as Lipschitz maps preserve the Hausdorff dimension, and the Hausdorff dimensions of $\mathbb{R}$ and $\mathbb{R}^2$ are 1 and 2 respectively).

- Theorem 2 is a *lower bound* – meaning we aim to exhibit an *example* of a hard instance when $G$ is not bijective. Having exhibited a hard instance for the $d_l = d_o = d$ case, it is trivially easy to construct a hard instance in the theorem where the output dimension is larger—which is the more common setting in practice. To do so, consider a circuit $\tilde{\mathcal{C}} : \{\pm 1\}^{d_l} \to \{\pm 1\}^{d_o}$, which equals to $\mathcal{C}$, the one-way-circuit from Conjecture 1 on the first $d_l$ coordinates, and is equal to 1 (i.e. is constant) on the last $d_o - d_l$ coordinates. Then $\tilde{\mathcal{C}}$ is one-way too—since the last $d_o - d_l$ values are just fixed to 1, if we can invert it, we can invert $\mathcal{C}$ too. The reduction in the proof of Theorem 2 then can just be performed with $\tilde{\mathcal{C}}$ instead, giving a lower bound instance for the $d_l < d_o$ case.

## 2.1 Upper bounds for bijective generators

We first lay out the assumptions on the map $G$. The first is a quantitative characterization of bijectivity; and the second requires upper bounds on the derivatives of $G$ up to order 3. We also have a centering assumption. We state these below.

**Assumption 1** (Strong invertibility). *We will assume that the latent and observable spaces have the same dimension (denoted $d$), and $G : \mathbb{R}^d \to \mathbb{R}^d$ is bijective. Moreover, we will assume there exists a positive constant $m > 0$ such that $\forall z_1, z_2 \in \mathbb{R}^d,\ \|G(z_1) - G(z_2)\| \geq m \cdot \|z_1 - z_2\|$.*

**Remark 2:** This is a stronger quantitative version of invertibility. Furthermore, the infinitesimal version of this condition (i.e. $\|z_1 - z_2\| \to 0$) implies that the smallest magnitude of the singular values of the Jacobian at any point is lower bounded by $m$, that is $\forall z \in \mathbb{R}^d,\ \min_{i \in [d]} |\sigma_i(J_G(z))| \geq m > 0$.

Since $m$ is strictly positive, this in particular means that the Jacobian is full rank everywhere.

**Remark 3:** Note, we do *not* require that $G$ is layerwise invertible (i.e. that the each map from one layer to the next is invertible) – if that is the case, at least in the limit $\beta \to 0$, the existence of an inference decoder of comparable size to $G$ is rather obvious: we simply invert each layer one at a time. This is important, as many architectures based on convolutions perform operations which increase the dimension (i.e. map from a lower to a higher dimensional space), followed by pooling (which decrease the dimension). Nevertheless, it has been observed that these architectures are invertible in

practice—Lipton & Tripathi (2017) manage to get almost 100% success at inverting an off-the-shelf trained model—thus justifying this assumption.

**Assumption 2** (Smoothness)**.** *There exists a finite positive constant $M > 0$ such that :*

$$\forall z_1, z_2 \in \mathbb{R}^d, \;\; \|G(z_1) - G(z_2)\| \leq M \cdot \|z_1 - z_2\|$$

*Moreover, we will assume that $G$ has continuous partial derivatives up to order 3 at every $z \in \mathbb{R}^d$ and the derivatives are bounded by finite positive constants $M_2$ and $M_3$ as*

$$\forall z \in \mathbb{R}^d, \;\; \left\|\nabla^2 G(z)\right\|_{op} \leq M_2 < \infty, \left\|\nabla^3 G(z)\right\|_{op} \leq M_3 < \infty$$

**Remark 4:** This is a mild assumption, stating that the map $G$ is smooth to third order. The infinitesimal version of this means that the largest magnitude of the singular values of the Jacobian at any point is upper bounded by $M$, that is $\forall z \in \mathbb{R}^d, \;\; \max_{i \in [d]} |\sigma_i(J_G(z))| = \|J_G(z)\|_{op} \leq M < \infty$.

**Remark 5:** A neural network with activation function $\sigma$ will satisfy this assumption when $\sigma : \mathbb{R} \to \mathbb{R}$ is Lipschitz, and $\max_a |\sigma'(a)|$ & $\max_a |\sigma''(a)|$ are finite.

**Assumption 3** (Centering)**.** *The map $G : \mathbb{R}^d \to \mathbb{R}^d$ satisfies $G(0) = 0$.*

**Remark 6:** This assumption is for convenience of stating the bounds—we effectively need the "range" of majority of the samples $x$ under the distribution of the generator. All the results can be easily restated by including a dependence on $\|G(0)\|$.

Our main result is then stated below. Throughout, the $\mathcal{O}(.)$ notation hides dependence on the map constants, namely $m, M, M_2, M_3$. We will denote by $d_{\text{TV}}(p, q)$ the total variation distance between the distributions $p, q$.

**Theorem 1** (Main, invertible generator)**.** *Consider a VAE generator given by a latent Gaussian with $\mu = 0$, $\Sigma = I$, noise parameter $\beta^2$ and generator $G : \mathbb{R}^d \to \mathbb{R}^d$ satisfying Assumptions 1 and 2, which has $N$ parameters and a differentiable activation function $\sigma$. Then, for*

$$\beta \leq \mathcal{O}\left(\frac{1}{d^{1.5}\sqrt{\log \frac{d}{\epsilon}}}\right) \tag{1}$$

*there exists a deep latent Gaussian with $N' = \mathcal{O}\left(N \cdot poly(d, \frac{1}{\beta}, \frac{1}{\epsilon})\right)$ parameters and activation functions $\{\sigma, \sigma', \rho\}$ and a neural network $\phi$ with $O(N')$ parameters and activation functions $\{\sigma, \sigma', \rho\}$, where $\rho(x) = x^2$, such that with probability $1 - \exp(-\mathcal{O}(d))$ over a sample $x$ from the VAE generator, $\phi(x)$ produces values for the parameters of the deep latent Gaussian such that the distribution $q(z|x)$ it specifies satisfies $d_{TV}(q(z|x), p(z|x)) \leq \epsilon$.*

**Remark 7:** Having an encoder $q(z|x)$ with parameters produced by a neural network taking $x$ as input is fairly standard in practice—it is known as amortized variational inference.(Mnih & Gregor, 2014; Kingma & Welling, 2013; Rezende et al., 2014)

**Remark 8:** The addition of $\rho$ in the activation functions is for convenience of stating the bound. Using usual techniques in universal approximation it can be simulated using any other smooth activation, see Appendix G.

## 2.2 LOWER BOUNDS FOR NON-BIJECTIVE GENERATORS

We now discuss the case when the generative map $G$ is not bijective, showing an instance such that no small encoder corresponding to the posterior exists. The lower bound will be based on a reduction from the existence of one-way functions—a standard complexity assumption in theoretical computer science (more concretely, cryptography). Precisely, we will start with the following form of the one-way-function conjecture:

**Conjecture 1** (Existence of one-way permutations (Katz & Lindell, 2020))**.** *There exists a bijection $f : \{-1, 1\}^d \to \{-1, 1\}^d$ computable by a Boolean circuit $\mathcal{C} : \{-1, 1\}^d \to \{-1, 1\}^d$ of size $poly(d)$, but for every $T(d) = poly(d)$ and $\epsilon(d) = \frac{1}{poly(d)}$ and circuit $\mathcal{C}' : \{-1, 1\}^d \to \{-1, 1\}^d$ of size $T(d)$ it holds $\mathbf{Pr}_{z \sim \{\pm 1\}^d}[\mathcal{C}'(\mathcal{C}(z)) = z] \leq \epsilon(d)$.*

In other words, there is a circuit of size polynomial in the input, s.t. for every polynomially sized invertor circuit (the two polynomials need not be the same—the invertor can be much larger, so long as it's polynomial), the invertor circuit succeeds on at most an inverse polynomial fraction of the inputs. Assuming this Conjecture, we show that there exist generators that do not have small encoders that accurately represent the posterior for most points $x$. Namely:

**Theorem 2** (Main, non-invertible generator). *If Conjecture 1 holds, there exists a VAE generator $G :$ $\mathbb{R}^d \to \mathbb{R}^d$ with size poly($d$) and activation functions $\{sgn, \min, \max\}$, s.t., for every $\beta = o(1/\sqrt{d})$, every $T(d) = poly(d)$ and every $\epsilon(d) = 1/poly(d)$, any encoder $E$ that can be represented by a deep latent Gaussian with networks that have total number of parameters bounded by $T(d)$, weights bounded by $W$, activation functions that are $L$-Lipschitz, and node outputs bounded by $M$ with probability $1 - \exp(-d)$ over a sample $x$ from $G$ and $L, M, W = o(\exp(poly(d)))$, we have:*

$$\mathbf{Pr}_{x \sim G} \left[ d_{TV}(E(z|x), p(z|x)) \leq \frac{1}{10} \right] \leq \epsilon(d)$$

Thus, we show the existence of a generator for which no encoder of polynomial size reasonably approximates the posterior for *even an inverse-polynomial* fraction of the samples $x$ (under the distribution of the generator).

**Remark 9:** The generator $G$, though mapping from $\mathbb{R}^d \to \mathbb{R}^d$ will be highly *non-invertible*. Perhaps counterintuitively, Conjecture 1 applies to bijections—though, the point will be that $G$ will be simulating a Boolean circuit, and in the process will give the same output on many inputs (more precisely, it will only depend on the sign of the inputs, rather than their values).

**Remark 10:** The choice of activation functions $\{sgn, \min, \max\}$ is for convenience of stating the theorem. Using standard universal approximation results, similar results can be stated with other activation functions, see Appendix G.

**Remark 11:** The restrictions on the Lipschitzness of the activations, bounds of the weights and node outputs of $E$ are extremely mild—as they are allowed to be potentially exponential in $d$—considering that even writing down a natural number in binary requires logarithmic number of digits.

## 3 RELATED WORK

On the empirical side, the impact of impoverished variational posteriors in VAEs (in particular, modeling the encoder as a Gaussian) has long been conjectured as one of the (several) reasons for the fuzzy nature of samples in trained VAEs. Zhao et al. (2017) provide recent evidence towards this conjecture. Invertibility of generative models in general (VAEs, GANs and normalizing flows), both as it relates to the hardness of fitting the model, and as it relates to the usefulness of having an invertible model, has been studied quite a bit: Lipton & Tripathi (2017) show that for off-the-shelf trained GANs, they can invert them with near-100% success rate, despite the model not being encouraged to be invertible during training; Dai & Wipf (2019) propose an alternate training algorithm for VAEs that tries to remedy algorithmic problems during training VAEs when data lies on a lower-dimensional manifold; Behrmann et al. (2020) show that trained normalizing flows, while being by design invertible, are just barely so — the learned models are extremely close to being singular.

On the theoretical side, the most closely relevant work is Lei et al. (2019). They provide an algorithm for inverting GAN generators *with random weights* and *expanding layers*. Their algorithm is layerwise — that is to say, each of the layers in their networks is invertible, and they invert the layers one at a time. This is distinctly not satisfied by architectures used in practice, which expand and shrink — a typical example are convolutional architectures based on convolutions and pooling. The same paper also shows NP-hardness of inverting a general GAN, but crucially they assume the network $G$ is part of the input (their proof does not work otherwise). Our lower bound can be viewed as a "non-uniform complexity" (i.e. circuit complexity) analogue of this, since we are looking for a small neural network $E$, as opposed to an efficient algorithm; crucially, however $G$ is *not* part of the input (i.e. $G$ can be preprocessed for an unlimited amount of time). Hand & Voroninski (2018) provide similar guarantees for inverting GAN generators with random weights that satisfy layerwise invertibility, albeit via non-convex optimization of a certain objective.

## 4 PROOF OVERVIEW

**Overview of Theorem 1**   Recall, given a generator $G$ and noise level $\beta^2$, we wish to provide a deep latent Gaussian (Definition 2) that approximates a sample from the distribution $p(z|x)$, with high probability over the draws of $x$ from the underlying density. To show Theorem 1, there are three main ingredients:

1. We show that gradient descent on the function $f(z) = \frac{1}{2}\|G(z) - x\|^2$, run for $\mathcal{O}(d/\beta^2)$ number of steps, recovers a point $z_{init}$ close to $\operatorname{argmax}_z \log p(z|x)$ – i.e. the mode of the posterior.

2. We show that Langevin diffusion, a Markov process that has $p(z|x)$ as it's stationary distribution, started close to the mode of the posterior via $z_{init}$ computed by GD above, run for $\mathcal{O}(\log(d/\epsilon))$ amount of time; returns a sample from a distribution close to $p(z|x)$ in total variation distance.

3. We show that the above two steps can be "simulated" by a deep latent Gaussian of size at most $poly(d, \frac{1}{\beta}, \frac{1}{\epsilon})$ larger than the original network, by inductively showing that each next iterate in the gradient descent and Markov chain can be represented by a neural network of size comparable to the generator $G$.

Part 1 is formalized in Lemma 1. The main observation is that the gradient of the loss will have the form $\nabla f(z) = J_G(z)^T(G(z) - x)$. By Strong Invertibility (Assumption 1), $J_G(z)$ will be an invertible matrix—therefore, any stationary points will necessarily be the (unique) point $z$ s.t. $G(z) = x$. This in turn, for small $\beta$, will be close to $\operatorname{argmax}_z \log p(z|x)$.

Part 2 is formalized in Lemma 2, and is technically most involved. As $\beta \to 0$, the distribution $p(z|x)$ concentrates more around it's mode, and approaches a Gaussian centered at it. More precisely, we show that starting at $z_{init}$, there is a region $\mathcal{D}$ such that the probability of Langevin diffusion leaving this region is small (Lemma 13) and the distribution restricted to $\mathcal{D}$ is log-concave (Lemma 4). As a consequence of the latter claim, reflected Langevin diffusion (a "restricted" version of the standard Langevin diffusion) converges rapidly to $p(z|x)$ restricted to $\mathcal{D}$. As a consequence of the former claim, the reflected and standard Langevin diffusion are close in the total variation sense.

Finally, Part 3 is based on the observation that we can approximately simulate gradient descent, as well as the Langevin dynamics using a deep latent Gaussian (Definition 2). Proceeding to gradient descent, the key observation is that if we give a neural network $z$ as the input, the point $z - \eta J_G(z)^T(G(z) - x)$ can be calculated by a network of size comparable to the size of a neural network representing $J_G$; the Jacobian of a neural network. And in turn, the Jacobian network can be represented by a neural network at most a factor $\mathcal{O}(d)$ bigger than the neural network representing $G$—this is by noting that each partial derivative of $G$ can be represented by a neural network of size comparable to $G$, essentially by the backpropagation algorithm (Lemma 5).

The idea to simulate Langevin dynamics is similar: the Euler-Mariyama discretization of the corresponding stochastic differential equation (with precision $h$) is a discrete-time Markov chain, s.t. the updates take the form

$$z \leftarrow z - h\nabla L(z) + \sqrt{2h}\xi_t, \ \ \xi \sim N(0, I) \tag{2}$$

where $L$ the unnormalized log-likelihood of the posterior $p(z|x)$ (equation 3 below). Same as is the case for gradient descent, a sample following the distribution of these updates can be implemented by a latent Gaussian, with a neural net of size at most $\mathcal{O}(d)$ bigger than the size of $G$—the randomness being handled by Gaussian noise that the latent Gaussian has its disposal.

**Overview of Theorem 2**   The proof of Theorem 2 proceeds by a reduction to Conjecture 1. We consider $G(z) = \mathcal{C}(\operatorname{sgn}(z))$, where $\mathcal{C}$ is a circuit satisfying the assumptions of Conjecture 1.

We show that we can use an encoder $q(z|x)$ that works with good probability over $x$ (wrt to the distribution of $G$) to construct a circuit $\mathcal{C}'$ that violates Conjecture 1. The main idea is as follows: the samples $x$ will, with high probability, lie near points on the hypercube $\{\pm 1\}^d$. For such points, the posterior $p(z|x)$ will concentrate around $\mathcal{C}^{-1}(\tilde{x})$, so taking the sign of a sample from $q(z|x)$ should, with good probability, produce an inverse of $\operatorname{sgn}(x)$.

Simulating this construction using a Boolean circuit, we can get a "randomized" circuit that inverts $\mathcal{C}$. Producing a deterministic circuit follows ideas from complexity theory, where we show that given a randomized circuit, there exists a single string of randomness that works for all inputs, if the success probability of the algorithm is high enough.

## 5 MAIN INGREDIENTS: THEOREM 1

First, we introduce a few pieces of notation that will be used in multiple places in the proofs. We will also omit subscripting quantities with $x$ to ease notation whenever $x$ is clear from context. For notational convenience, we will denote the unnormalized log-likelihood of $p(z|x)$ as $L : \mathbb{R}^d \to \mathbb{R}$, s.t.

$$L(z) := \frac{1}{2}\left(\|z\|^2 + \frac{1}{\beta^2}\|G(z) - x\|^2\right) \tag{3}$$

Note, by Bayes rule, $p(z|x) \propto p(x|z)\, p(z)$, so $p(z|x) \propto e^{-L(z)}$. Note this is a well-defined probability distribution since $\forall z \in \mathbb{R}^d, \quad L(z) \geq \frac{1}{2}\|z\|^2$, therefore the partition function is finite. Further, we will denote the inverse of $x$ as $\hat{z}$, namely:

$$\hat{z} := \arg\min_{z \in \mathbb{R}^d} \|G(z) - x\|^2 \tag{4}$$

Note since $G$ is invertible, $\hat{z}$ is uniquely defined, and it's such that $G(\hat{z}) = x$. Crucially due to this, the function $L(z)$ is strongly convex in some region around the point $\hat{z}$ – which can be seen by evaluating $\nabla^2 L(\hat{z})$ from equation 3. These properties are convenient – that is why we base our analysis on this inverse point.

### 5.1 PART 1: CONVERGENCE OF GRADIENT DESCENT

First, we show that gradient descent on the function $f(z) = \frac{1}{2}\|G(z) - x\|^2$, started at the origin, converges to a point close to $\hat{z}$ in a bounded number of steps. Precisely:

**Lemma 1** (Convergence of Gradient Descent to the Inverse). *When $G$ satisfies Assumptions 1 and 2, with probability $1 - \exp(-\mathcal{O}(d))$ over the choice of $x$, running gradient descent on $f(z) = \frac{1}{2}\|G(z) - x\|^2$ with the starting point $z^{(0)} := 0$ and learning rate $\eta := \frac{1}{Q}$ where $Q = \mathcal{O}(d)$, for $S = \mathcal{O}(\frac{d^2}{\delta^2})$ steps converges to a point $z^{(S)}$, such that $\|z^{(S)} - \hat{z}\| \leq \delta$.*

As mentioned in Section 4, the idea is that while $f$ is non-convex, its unique stationary point is $\hat{z}$, as $\nabla f(z) = J_G(z)^T (G(z) - x)$ and $J_G(z)$ is an invertible matrix $\forall z \in \mathbb{R}^d$ from Assumption 1. The proof essentially proceeds by a slight tweak on the usual descent lemma (Nesterov, 2003), due to the fact that the Hessian's norm is not bounded in all of $\mathbb{R}^d$ and is given in Appendix B.

### 5.2 PART 2: FAST MIXING OF LANGEVIN DIFFUSION

We first review some basic definitions and notation. Recall, Langevin diffusion is described by a stochastic differential equation (SDE):

**Definition 3** (Langevin Diffusion). Langevin Diffusion is a continuous-time Markov process $\{z_t : t \in \mathbb{R}^+\}$ defined by the stochastic differential equation

$$dz_t = -\nabla g(z_t)dt + \sqrt{2}dB_t \qquad t \geq 0 \tag{5}$$

where $g : \mathbb{R}^d \to \mathbb{R}$, and $\{B_t : t \in \mathbb{R}^+\}$ is a Brownian motion in $\mathbb{R}^d$ with covariance matrix $I_d$. Under mild regularity conditions, the stationary distribution of this process is $P : \mathbb{R}^d \to \mathbb{R}^+$ such that $P(z) \propto e^{-g(z)}$.

For the reader unfamiliar with SDEs, the above process can be interpreted as the limit as $h \to 0$ of the Markov chain

$$z \leftarrow z - h\nabla L(z)dt + \sqrt{2h}\xi, \quad \xi \sim N(0, I) \tag{6}$$

The convergence time of the chain to stationarity is closely tied to the convexity properties of $L$, through functional-analytic tools like Poincaré inequalities and Log-Sobolev inequalities (Bakry & Émery, 1985). We show that the distribution $p(z|x)$ can be sampled $\epsilon$-approximately in total variation distance by running Langevin dynamics for $\mathcal{O}(\log(d/\epsilon))$ number of steps. Namely:

**Lemma 2** (Sampling Complexity for general $G$). *Let $\beta = \mathcal{O}\left(\frac{1}{d^{1.5}\sqrt{\log \frac{d}{\epsilon}}}\right)$. Then, with probability $1 - \exp(-\mathcal{O}(d))$ over the choice of $x$, if we initialize Langevin diffusion equation 6 at $z_0$ which satisfies $\|z_0 - \hat{z}\| \leq \Omega\left(\beta \cdot \sqrt{d \log \frac{d}{\epsilon}}\right)$, for some $T = \mathcal{O}(\frac{1}{d^2}\log \frac{1}{\epsilon})$ we have $d_{TV}(P_T, p(z|x)) \leq \epsilon/2$ where $P_t$ is the density corresponding to $z_t$.*

Note, the lemma requires the variance of the latent Gaussian $\beta$ to be sufficiently small. This intuitively ensures the posterior $p(z|x)$ is somewhat concentrated around its mode — and we will show that when this happens, in some region around the mode, the posterior is close to a Gaussian. The proof of this has two main ingredients: (1) $z_t$ stays in a region $\mathcal{D}$ around $\hat{z}$ with high probability; (2) The distribution $p(z|x)$, when restricted to $\mathcal{D}$ is log-concave, so a "projected" version of Langevin diffusion (formally called *reflected Langevin diffusion*, see Definition 4) mixes fast to this restricted version of $p(z|x)$.

The first part is formalized as Lemma 13 in Appendix C. The main mathematical tool is a stochastic differential equation characterizing the change of the distance from $\hat{z}$. Namely, we show:

**Lemma 3** (Change of distance from $\hat{z}$). *Let $\eta(z) := \frac{1}{2}\|z - \hat{z}\|^2$. If $z$ follows the Langevin SDE corresponding to $z_t$, then for $z \in \mathcal{D}$ as defined in Lemma 13 and $\beta \leq \beta_0$ as defined above, it holds that*

$$d\eta(z) \leq -\frac{m^2}{\beta^2} \cdot \eta(z)dt + \left(\|\hat{z}\|^2 + d\right)dt + 2\sqrt{\eta(z)}dB_t$$

This SDE is analyzed through the theory of Cox-Ingersoll-Ross (Cox et al., 2005) processes, which are SDEs of the type $dM_t = (a - bM_t)dt + c\sqrt{M_t}dB_t$, where $a, b, c$ are positive constants. Intuitively, this SDE includes an 'attraction term' $-bM_t dt$ which becomes stronger as we increase $M_t$ (in our case, as we move away from the center of the region $\hat{z}$), and a diffusion term $c\sqrt{M_t}dB_t$ due to the injection of noise. The term $(\|\hat{z}\|^2 + d)dt$ comes from the heuristic Itó calculus equality $dB_t^2 = dt$. The "attraction term" $-\frac{m^2}{\beta^2}\eta(z)dt$ comes from the fact that $L(z)$ is approximately strongly convex near $\hat{z}$. (Intuitively, this means the Langevin process is attracted towards $\hat{z}$.)

To handle the second part, we show the distribution $p(z|x)$ restricted to $\mathcal{D}$ is log-concave:

**Lemma 4** (Strong convexity of $L$ over $\mathcal{D}$). *For all $z \in \mathcal{D}$, $\nabla^2 L(z) \succeq I$.*

Since the set $\mathcal{D}$ is a $l_2$ ball (which is of course convex), we can show an alternate Markov process which always remains inside $\mathcal{D}$, called the *reflected Langevin diffusion* mixes fast.

**Definition 4** (Reflected Langevin diffusion, (Lions & Sznitman, 1984; Saisho, 1987)). For a sufficiently regular region $\mathcal{D} \subseteq \mathbb{R}^d$, there exists a measure $U$ supported on $\partial \mathcal{D}$, such that the continuous-time diffusion process $\{y_t : t \in \mathbb{R}^+\}$ starting with $y_0 \in \mathcal{D}$ defined by the stochastic differential equation:

$$dy_t = -\nabla g(y_t)dt + \sqrt{2}dB_t + \nu_t U(y_t)dt \qquad t \geq 0 \tag{7}$$

has as stationary measure $\tilde{P} : \mathcal{D} \to \mathbb{R}^+$ such that $\tilde{P}(y) \propto e^{-g(y)}$. Here $g : \mathbb{R}^d \to \mathbb{R}$, $\{B_t : t \in \mathbb{R}^+\}$ is a Brownian motion in $\mathbb{R}^d$ with covariance matrix $I_d$, and $\nu_t$ is an outer normal unit vector to $\mathcal{D}$.

For readers unfamiliar with SDEs, the above process can be interpreted as the limit as $h \to 0$ of

$$\forall k \geq 0, \ \ y_{k+1} = \Pi_{\mathcal{D}}\left(y_k - h\nabla g(y_k) + \sqrt{2h}\xi_k\right) \qquad \xi_k \sim \mathcal{N}(0, I_d) \tag{8}$$

where $\Pi_{\mathcal{D}}$ is the projection onto the set $\mathcal{D}$.

For total variationa distance, the mixing time in total variation distance of this diffusion process was characterized by Bubeck et al. (2018):

**Theorem 3** (Mixing for log-concave distributions over convex set, Bubeck et al. (2018): Proposition 2). *Suppose a measure $p : \mathbb{R}^d \to \mathbb{R}^+$ of the form $p(x) \propto e^{-g(x)}$ is supported over $S \subseteq \mathbb{R}^d$ of diameter $R$ which is convex, and $\forall x \in S, \nabla^2 g \gtrsim 0$. Furthermore, let $x_0 \in S$. Then, if $p_t$ is the distribution after running equation 7 for time $t$ starting with a Dirac distribution at $x_0$, we have*

$$TV(p_t, p) \lesssim e^{-\frac{t}{2R^2}}$$

Since the set $\mathcal{D}$ in Lemma 13 is an $l_2$ ball (which is of course convex), it suffices to show that $L$ is convex on $\mathcal{D}$—which is a straightforward calculation. The details of this are again in Appendix C.

### 5.3 PART 3: SIMULATION BY NEURAL NETWORKS

Finally, we show that the operations needed for gradient descent as well as Langevin sampling can be implemented by a deep latent Gaussian. Turning the gradient descent first, we need to "simulate" by a

neural net the operation $z \leftarrow z - \eta J_G(z)^T (G(z) - x)$ roughly $\mathcal{O}(d/\beta^2)$ number of times. We prove that there is a neural network that given an input $z$, outputs $z - \eta J_G(z)^T (G(z) - x)$. The main ideas for this are that: (1) If $G$ is a neural network with $N$ parameters, each coordinate of $J_G(z)$ can be represented as a neural network with $\mathcal{O}(N)$ parameters. (2) If $f, g$ are two neural networks with $N$ and $M$ parameters respectively, $f + g$ and $fg$ can be written as neural networks with $\mathcal{O}(M + N)$ parameters.

The first part essentially follows by implementing backpropagation with a neural network:

**Lemma 5** (Backpropagation Lemma, Rumelhart et al. (1986)). *Given a neural network $G : \mathbb{R}^d \to \mathbb{R}$ of depth $l$ and size $N$, there is a neural network of size $\mathcal{O}(N + l)$ which calculates the gradient $\partial G/\partial z_i$ for $i \in [d]$.*

For the second part, addition is trivially implemented by "joining" the two neural networks and adding one extra node to calculate the sum of their output. For multiplication, the claim follows by noting that $xy = \frac{1}{4}(x + y)^2 - \frac{1}{4}(x - y)^2$. Using the square activation function, this can be easily computed by a 2-layer network.

Proceeding to the part about simulating Langevin diffusion, we first proceed to create a discrete-time random process that closely tracks the diffusion. The intuition for this was already mentioned in Section 4— the SDE describing Langevin can be viewed as the limit $h \to 0$ of

$$z \leftarrow z - h \left[ z + \frac{1}{\beta^2} J_G(z)^T (G(z) - x) \right] + \sqrt{2h}\xi, \xi \sim N(0, I)$$

It can be readily seen that the drift part, that is $\frac{1}{\beta^2} J_G(z)^T (G(z) - x)$ can be calculated by a neural net via the same token as in the case of the gradient updates. Thus, the only difference is in the injection of Gaussian noise. However, each layer $G_i$ in a deep latent Gaussian exactly takes a Gaussian sample $\xi$ and outputs $G_i(z) + \gamma\xi$ – so each step of the above discretization of Langevin can be exactly implement by a layer in a deep latent Gaussian. The details are in Appendix D.

## 6 Main ingredients: Theorem 2

We prove Theorem 2 proceeds by a reduction to Conjecture 1. Namely, if $\mathcal{C}$ is a Boolean circuit satisfying the one-way-function assumptions in Conjecture 1, we consider $G(z) = \mathcal{C}(\text{sgn}(z))$. Suppose there is a deep latent Gaussian of size $T(n)$ with corresponding distribution $q(z|x)$, s.t. with probability at least $\epsilon(d)$ over the choice of $x$ (wrt to the distribution of $G$), it satisfies $d_{\text{TV}}(p(z|x), q(z|x)) \leq 1/10$. We will how to construct from the encoder a circuit $\mathcal{C}'$ that violates Conjecture 1. Namely, consider a $\tilde{x}$ constructed by sampling $\tilde{z} \sim \{\pm 1\}^d$ uniformly at random, and outputting $\tilde{x} = \mathcal{C}(\tilde{z})$. To violate Conjecture 1, we'd like to find $\mathcal{C}^{-1}(\tilde{x})$, using the encoder $E$.

We first note that $x = \tilde{x} + \xi$, $\xi \sim N(0, \beta^2 I_d)$ is distributed according to the distribution of the generator $G$. Moreover, we can show that with probability at least $\epsilon(d)/2$, it will be the case that $x$ is s.t. both $d_{\text{TV}}(p(z|x), q(z|x)) \leq 1/10$ and $x$ is very close to a point on the discrete hypercube $\{\pm 1\}^d$. The reason for this is that the samples $x$ themselves for small enough $\beta$ with high probability are close to a point on the hypercube. For such points, it will be the case that $q(z|x)$ is highly concentrated around $\mathcal{C}^{-1}(\tilde{x})$, thus returning the sign of a sample will with high probability invert $\tilde{x}$. By simulating this process using a Boolean circuit, we get a "randomized" circuit (that is, a circuit using randomness) that succeeds at inverting $\mathcal{C}$ with inverse polynomial success. To finally produce a *deterministic* Boolean circuit that violates Conjecture 1, we use a standard trick in complexity theory: boosting the success probability by generating multiple samples together with the assumption on the weights and Lipschitzness of $E$. For details, refer to Appendix E.

## 7 Conclusion

In this paper we initiated the first formal study of the representational complexity of the good encoders in VAEs. We proved the following dichotomy: generators with invertible means give rise to posteriors that can be approximated by an encoders not much larger than the generator. On the other hand, for non-invertible generators, the corresponding encoder may need to be exponentially larger. We hope our work will stimulate research of other data distribution properties that render inference tractable.

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

## A    Technical Definitions

In this section, we state several definitions and standard lemmas, to cover some technical prerequisites for the proof.

We first recall a few definitions and lemmas about two notions of distance between probabilities we will use extensively, total variation and $\chi^2$ distance.

**Definition 5** (Total variation distance). The total variation distance between two distributions $P, Q$ is defined as

$$d_{\mathbf{TV}}(P, Q) := \sup_A |P(A) - Q(A)|$$

**Definition 6** ($\chi^2$ distance). The $\chi^2$ distance between two distributions $P, Q$, s.t. $P$ is absolutely continuous with respect to $Q$, is defined as

$$\chi^2(P, Q) := \mathbb{E}_Q \left( \frac{P}{Q} - 1 \right)^2$$

The following two lemmas about the total variation and $\chi^2$ distance are standard:

**Lemma 6** (Coupling Lemma). *Let $P, Q : \Omega \to \mathbb{R}$ be two distributions, and $c : \Omega^{\otimes 2} \to \mathbb{R}$ be any coupling of $P, Q$. Then, if $(X, X')$ are random variables following the distribution $c$, we have:*

$$d_{TV}(P, Q) \leq \mathbf{Pr}[X \neq X']$$

**Lemma 7** (Inequality between TV and $\chi^2$). *Let $P, Q$ be probability measures, s.t. $P$ is absolutely continuous with respect to $Q$. We then have:*

$$d_{TV}(P, Q) \leq \frac{1}{2} \sqrt{\chi^2(P, Q)}$$

We will be heavily using notions from continuous-time Markov Processes. In particular:

**Definition 7** (Markov semigroup). We say that a family of functions $\{P_t(x, y)\}_{t \geq 0}$ on a state space $\Omega$ is a Markov semigroup if $P_t(x, \cdot)$ is a distribution on $\Omega$ and

$$\mathbf{Pr}_{t+s}(x, y) = \int_\Omega P_t(x, z) P_s(z, y) dz$$

for all $x, y \in \Omega$ and $s, t \geq 0$.

**Definition 8** (Continuous time Markov processes). A continuous time Markov process $(X_t)_{t \geq 0}$ on state space $\Omega$ is defined by a Markov semigroup $\{P_t(x, y)\}_{t \geq 0}$ as follows. For any measurable $A \subseteq \Omega$

$$\mathbf{Pr}(X_{s+t} \in A) = \int_A P_t(x, y) dy := P_t(x, A)$$

Moreover $P_t$ can be thought of as acting on a function $g$ as

$$(P_t g)(x) = \mathbb{E}_{P_t(x, \cdot)}[g(y)] = \int_\Omega g(y) P_t(x, y) dy$$

Finally we say that $p(x)$ is a stationary distribution if $X_0 \sim p$ implies that $X_t \sim p$ for all $t$.

With this in place, we can finally define the Langevin diffusion Markov process:

**Definition 9** (Langevin Diffusion, Definition 3 restated). Langevin Diffusion is a continuous-time Markov process $\{z_t : t \in \mathbb{R}^+\}$ defined by the stochastic differential equation

$$dz_t = -\nabla g(z_t) dt + \sqrt{2} dB_t \qquad t \geq 0 \tag{9}$$

where $g : \mathbb{R}^d \to \mathbb{R}$, and $\{B_t : t \in \mathbb{R}^+\}$ is a Brownian motion in $\mathbb{R}^d$ with covariance matrix $I_d$. Under mild regularity conditions, the stationary distribution of this process is $P : \mathbb{R}^d \to \mathbb{R}^+$ such that $P(z) \propto e^{-g(z)}$.

It will also be useful to have a definition of a diffusion process that only stays within some support $\mathcal{D}$, and thus has a stationary distribution supported on $\mathcal{D}$. Specifically:

**Definition 10** (Reflected Langevin diffusion, (Lions & Sznitman, 1984; Saisho, 1987), Definition 4 restated). For a sufficiently regular region $\mathcal{D} \subseteq \mathbb{R}^d$, there exists a measure $U$ supported on $\partial\mathcal{D}$, such that the continuous-time diffusion process $\{y_t : t \in \mathbb{R}^+\}$ starting with $y_0 \in \mathcal{D}$ defined by the stochastic differential equation:

$$dy_t = -\nabla g(y_t)dt + \sqrt{2}dB_t + \nu_t U(y_t)dt \qquad t \geq 0 \tag{10}$$

has as stationary measure $\tilde{P} : \mathcal{D} \to \mathbb{R}^+$ such that $\tilde{P}(y) \propto e^{-g(y)}$. Here $g : \mathbb{R}^d \to \mathbb{R}$, $\{B_t : t \in \mathbb{R}^+\}$ is a Brownian motion in $\mathbb{R}^d$ with covariance matrix $I_d$, and $\nu_t$ is an outer normal unit vector to $\mathcal{D}$.

Towards introducing quantities governing the mixing time of a continuous-time Markov process, we introduce

**Definition 11.** The generator $\mathcal{L}$ of the Markov Process is defined (for appropriately restricted functionals $g$) as

$$\mathcal{L}g = \lim_{t \to 0} \frac{P_t g - g}{t}.$$

Moreover if $p$ is the unique stationary distribution, the corresponding Dirichlet form and variance are

$$\mathcal{E}_M(g, h) = -\mathbb{E}_p\langle g, \mathcal{L}h \rangle \text{ and } \mathrm{Var}_p(g) = \mathbb{E}_p(g - \mathbb{E}_p g)^2$$

respectively. We will use the shorthand $\mathcal{E}(g) := \mathcal{E}(g, g)$.

A quantity of particular interest when analyzing the mixing time of Markov processes is the Poincaré constant:

**Definition 12** (Poincaré Inequality). A continuous Markov process satisfies a Poincaré inequality with constant $C$ if for all functions $g$ such that $\mathcal{E}_M(g)$ is defined (finite), it holds that:

$$\mathcal{E}_M(g) \geq \frac{1}{C} \cdot \mathrm{Var}_P(g)$$

where $P$ (in the subscript) is the stationary distribution of the Markov process. We will abuse notation, and for such a Markov process with stationary distribution $P$, denote by $\mathbf{C_{pc}}(P)$ the smallest $C$ such that above Poincaré inequality is satisfied; and call it the *Poincaré constant of $P$*.

The relationship between the Poincaré constant and the mixing time of a random walk (in particular, the one in Definition 4) is given by the following (standard) lemma:

**Lemma 8** (Mixing in $\chi^2$ from Poincaré). *Let $\tilde{z}_t$ follow the stochastic differential equation of reflected Langevin diffusion (Definition 4) with stationary measure $\tilde{P}$. Let $\tilde{P}_t$ be the density of $\tilde{z}_t$ and let $\tilde{P}_0$ be absolutely continuous with respect to the Lebesgue measure. Then:*

*(1) If $\tilde{P}_0$ is supported on $\mathcal{D}$, $\tilde{P}_t$ is supported on $\mathcal{D}$, $\forall t > 0$*

*(2) $\chi^2(\tilde{P}_t, \tilde{P}) \leq e^{-t/\mathbf{C_{pc}}}\chi^2(\tilde{P}_0, \tilde{P})$*

Finally, we will also need a discretized version of the continuous-time Langevin diffusion.

**Definition 13** (Euler-Mariyama Discretization for the Langevin Diffusion). For a potential $g : \mathbb{R}^d \to \mathbb{R}$, the Euler-Mariyama discretization of the continuous Langevin diffusion with discretization level $h$ is the discrete-time random walk

$$\forall k \geq 0, \ \ z_{k+1} = z_k - h\nabla g(z_k) + \sqrt{2h}\xi_k \qquad \xi_k \sim \mathcal{N}(0, I_d) \tag{11}$$

We will also associate with this discrete-time process a stochastic differential equation

$$d\hat{z}_t = -\nabla g(\hat{z}_{\lfloor t/h \rfloor h})dt + \sqrt{2}dB_t \tag{12}$$

which has the property that $\hat{z}_{kh}$ and $z_k$ follow the same distribution for all $k \geq 0$, if $z_0$ and $\hat{z}_0$ follow the same distribution.

Note the equivalence between equation 11 and equation 12 follows by noting that the drift in the interval $[th, (t+1)h]$ in equation 12 is constant – so we only need to integrate the Brownian motion for time $h$.

We can also consider the discretized version of the reflected Langevin diffusion, which looks like projected gradient descent with Gaussian noise Bubeck et al. (2018).

**Definition 14** (Discretization for the Reflected Langevin Diffusion). For a potential $g : \mathbb{R}^d \to \mathbb{R}$, the Euler-Mariyama discretization of the reflected Langevin diffusion with discretization level $h$ is the discrete-time random walk

$$\forall k \geq 0, \ \ y_{k+1} = \Pi_{\mathcal{D}} \left( y_k - h\nabla g(y_k) + \sqrt{2h}\xi_k \right) \qquad \xi_k \sim \mathcal{N}(0, I_d) \tag{13}$$

where $\Pi_{\mathcal{D}}$ is the projection onto the set $\mathcal{D}$.

We can also associate with this discrete-time process a stochastic differential equation

$$d\hat{y}_t = -\nabla g(\hat{y}_{\lfloor t/h \rfloor h})dt + \sqrt{2}dB_t + \nu_t U(\hat{y}_t)dt \tag{14}$$

for a measure $U$ supported on $\partial\mathcal{D}$, which has the property that $\hat{y}_{kh}$ and $y_k$ follow the same distribution for all $k \geq 0$, if $y_0$ and $\hat{y}_0$ follow the same distribution.

## B    PROOF OF LEMMA 1: CONVERGENCE OF GRADIENT DESCENT TO $\hat{z}$

First, we state the tweak on the usual descent lemma (Nesterov, 2003), due to the fact that the Hessian's norm is not bounded in all of $\mathbb{R}^d$. We only need to identify a region $\mathcal{A}$ containing the starting point $z^{(0)}$ such that the function's value at points outside $\mathcal{A}$ are guaranteed to be more than the function value at $z^{(0)}$. Namely:

**Lemma 9** (Adjusted Descent Lemma). *Let $f : \mathbb{R}^d \to \mathbb{R}$ be a twice differentiable function. For a region $\mathcal{A}$, let it hold that $\forall z \in \mathcal{A}, \ \|\nabla^2 f(z)\|_{op} \leq Q < \infty$. Further, let $\forall z \in \mathbb{R}^d \backslash \mathcal{A}, \ f(z) \geq f(z^{(0)})$ for a known point $z^{(0)} \in \mathcal{A}$. Then the gradient descent iterates starting with $z^{(0)}$ with the learning rate $\eta := \frac{1}{Q}$ satisfy*

$$f(z^{(s+1)}) \leq f(z^{(s)}) - \frac{1}{2Q}\|\nabla f(z^{(s)})\|^2 \qquad s \geq 0$$

We first state the well-known descent lemma.

**Lemma 10** (Descent Lemma, Eq 1.2.13 in (Nesterov, 2003)). *Let $f$ be a twice differentiable function, and at a point $z$, the hessian satisfies $\|\nabla^2 f(z)\|_{op} \leq Q < \infty$. Then with the learning rate $\eta := \frac{1}{Q}$, the next gradient descent iterate $z^+$ satisfies*

$$f(z^+) \leq f(z) - \frac{1}{2Q}\|\nabla f(z)\|^2$$

Using this, we prove the Adjusted Descent Lemma.

*Proof of Lemma 9.* We will prove the Lemma by induction. To that end, let us call by $I(s), \forall s \geq 0$ the proposition in the Lemma statement, i.e.

$$I(s): \qquad f(z^{(s+1)}) \leq f(z^{(s)}) - \frac{1}{2Q}\|\nabla f(z^{(s)})\|^2$$

We first write two more propositions $J(s)$ and $K(s)$:

$$J(s): \qquad z^{(s)} \in \mathcal{A}$$
$$K(s): \qquad f(z^{(s)}) \leq f(z) \ \ \forall z \in \mathbb{R}^d \setminus \mathcal{A}$$

We will use induction to show that $I(s) \wedge J(s) \wedge K(s)$ hold true $\forall s \geq 0$. $J(0)$ holds true, since $z^{(0)} \in \mathcal{A}$ is given in the statement. $K(0)$ is also given in the statement of the lemma.

Since $\forall z \in \mathcal{A}, \ \|\nabla^2 f(z)\|_{op} \leq Q$; from Lemma 10, we conclude that

$$J(s) \implies I(s) \qquad \forall s \geq 0 \tag{15}$$

Hence $I(0)$ also holds true. So the base case of the induction ($s = 0$) is done, i.e. $I(0) \wedge J(0) \wedge K(0)$.

Since $I(s)$ implies descent, and $K(s)$ implies that all points outside region $\mathcal{A}$ have larger function value, we also have

$$I(s) \wedge K(s) \implies J(s+1) \wedge K(s+1) \qquad \forall s \geq 0 \tag{16}$$

Combining implications equation 15 and equation 16 and Lemma 10, we get

$$I(s) \wedge J(s) \wedge K(s) \implies I(s+1) \wedge J(s+1) \wedge K(s+1) \qquad \forall s \geq 0 \tag{17}$$

Hence the induction step also goes through. $\qquad\square$

We can now use this to prove convergence of gradient descent.

*Proof of Lemma 1.* To use the adjusted descent lemma (Lemma 9), let us define the appropriate region $\mathcal{A}$.

$$\mathcal{A} := \left\{ z : \|z\| \leq \left( \frac{M}{m} + 1 \right) \cdot \frac{\|x\|}{m} \right\} \tag{18}$$

Obviously $z^{(0)} = 0 \in \mathcal{A}$. We also have $\forall z \in \mathbb{R}^d \setminus \mathcal{A}$, the following holds

$$
\begin{aligned}
f(z) - f(z^{(0)}) &= f(z) - f(0) \\
&= \frac{1}{2} \left( \|G(z) - x\|^2 - \|G(0) - x\|^2 \right) \\
&= \frac{1}{2} \left( \|G(z) - G(\hat{z})\|^2 - \|G(0) - G(\hat{z})\|^2 \right) \\
&\geq^{(0)} \frac{1}{2} \left( m^2 \|z - \hat{z}\|^2 - M^2 \|\hat{z}\|^2 \right) \\
&\geq C_+ \left( m \|z - \hat{z}\| - M \|\hat{z}\| \right) \\
&\geq m C_+ \left( \|z - \hat{z}\| - \frac{M}{m} \cdot \|\hat{z}\| \right) \\
&\geq m C_+ \left( \|z\| - \left( \frac{M}{m} + 1 \right) \cdot \|\hat{z}\| \right) \\
&\geq^{(1)} m C_+ \left( \left( \frac{M}{m} + 1 \right) \cdot \frac{\|x\|}{m} - \left( \frac{M}{m} + 1 \right) \cdot \|\hat{z}\| \right) \\
&\geq \left( \frac{M}{m} + 1 \right) \cdot m C_+ \cdot \left( \frac{\|x\|}{m} - \|\hat{z}\| \right) \\
&\geq^{(2)} 0
\end{aligned}
$$

$C_+ = \frac{1}{2} \left( m \|z - \hat{z}\| + M \|\hat{z}\| \right)$ denotes a non-negative quantity, which does not affect the sign of these inequalities. In the above calculation, $(0)$ follows from Assumptions 1 and 2, $(1)$ uses the definition of $\mathcal{A}$ from equation equation 18, and $(2)$ uses Lemma 23.

In $\mathcal{A}$, the norm of the Hessian can be bounded below. The expression of $\nabla^2 f$ can be inferred from the expression of $\nabla^2 L$ given in equation equation 56. We then get

$$
\begin{aligned}
\left\| \nabla^2 f(z) \right\|_{op} &\leq \|J_G(z)\|_{op}^2 + \sum_{i \in [d]} |G_i(z) - x_i| \cdot \left\| \nabla^2 G_i(z) \right\|_{op} \\
&\leq^{(0)} M^2 + \left( \sum_{i \in [d]} |G_i(z) - x_i| \right) \cdot \max_{i \in [d]} \left\| \nabla^2 G_i(z) \right\|_{op} \\
&\leq^{(1)} M^2 + \sqrt{d} \cdot \|G(z) - x\| \cdot M_2 \\
&\leq M^2 + \sqrt{d} M M_2 \cdot \|z - \hat{z}\| \\
&\leq^{(2)} M^2 + \sqrt{d} M M_2 \cdot 2 \left( \frac{M}{m} + 1 \right) \cdot \frac{\|x\|}{m} =: Q \qquad z \in \mathcal{A} \tag{19}
\end{aligned}
$$

where $(0)$ follows from Assumption 2, $(1)$ follows since $\forall i \in [d], \left\| \nabla^2 G_i(z) \right\|_{op} \leq \|\nabla^2 G(z)\|_{op}$; which follows from the sup-definition of the $\ell_2$ norm (Lim, 2005). In $(1)$, we also used the fact that $\|v\|_1 \leq \sqrt{\dim(v)} \cdot \|v\|_2$. And in $(2)$ we used the diameter of $\mathcal{A}$ of equation equation 18.

With $\mathcal{A}, Q$ defined through equations equation 18 and equation 19, the conditions of Lemma 9 are satisfied, so we can use the descent equation. We can then use the standard argument below

$$\forall s \in [0, S], \ \left\| \nabla f(z^{(s)}) \right\| \geq \alpha$$

$$\implies f(z^{(s)}) \leq f(z^{(0)}) - S \cdot \frac{1}{2Q} \cdot \alpha^2$$

Since $f$ is a non-negative function, it is bounded below by zero. This means that

$$\min_{s \in [0,S]} \|\nabla f(z^{(s)})\| \leq \sqrt{\frac{2Q \cdot f(z^{(0)})}{S}}$$

We note that

$$\begin{aligned}
\|\nabla f(z)\| &= \|J_G(z)^T \left( G(z) - x \right)\| \\
&\geq \min_{i \in [d]} |\sigma_i(J_G(z))| \, \|G(z) - x\| \\
&\geq \min_{i \in [d]} |\sigma_i(J_G(z))| \, \sqrt{2f(z)} \\
&\geq m \sqrt{2f(z)}
\end{aligned}$$

By Lemma 9 we then have

$$\begin{aligned}
f(z^{(S)}) &\leq \frac{1}{2} \left( \frac{\min_{s \in [0,S]} \|\nabla f(z^{(s)})\|}{m} \right)^2 \\
&\leq \frac{1}{m^2} \frac{Q f(z^{(0)})}{S}
\end{aligned}$$

Furthermore, by Assumption 1 we have

$$\begin{aligned}
\|z^{(S)} - \hat{z}\| &\leq \frac{1}{m} \|G(z^{(S)}) - x\| \\
&= \frac{1}{m} \sqrt{2f(z^{(S)})} \\
&\leq \frac{1}{m^2} \sqrt{\frac{2Q f(z^{(0)})}{S}}
\end{aligned}$$

Hence, setting $\delta = \frac{1}{m^2} \cdot \sqrt{\frac{2Q f(z^{(0)})}{S}}$ gives us the required number of gradient descent steps. We will finally simplify the expression for $S$. Since $f(z^{(0)}) = f(0) = \frac{1}{2}\|G(0) - x\|^2 = \frac{1}{2}\|x\|^2$ (from Assumption 3), we get

$$S = \frac{Q \cdot \|x\|^2}{m^4} \cdot \frac{1}{\delta^2} \tag{20}$$

Plugging in the expression for $Q$ from equation 19, and using $\|x\| = \mathcal{O}(\sqrt{d})$ from Lemma 24, we get $S = \mathcal{O}(\frac{d^2}{\delta^2})$ as desired. $\qquad\square$

## C    Proof of Lemma 2: Fast Mixing of Langevin Diffusion

**Notation:**    We first set up some notation. We denote by $z_t$ the time $t$ iterate of Langevin diffusion (Definition 3) with $g := L$, having as stationary distribution $P(z) \equiv p(z|x)$. Further, we denote by $P_t$ the density of $z_t$.

Similarly, we denote by $\tilde{z}_t$ the iterates of the reflected Langevin chain (Definition 4) with $g := L$, with stationary distribution $\tilde{P}(z) \propto e^{-L(z)}$ supported over $z \in \mathcal{D}$. We denote by $\tilde{P}_t$ the density of $\tilde{z}_t$.

**Lemma 11.** $rad(\mathcal{D})$ *defined in Lemma 13 along with the restriction on $\beta$ from the same lemma satisfies*

$$rad(\mathcal{D}) \leq \min \left\{ \frac{m^2}{6dMM_2}, \frac{m}{\sqrt{2\sqrt{d}MM_3}} \right\}$$

*This in particular means that $rad(\mathcal{D}) \leq \mathcal{O}(1/d)$.*

*Proof.* Follows by straightforward substitution. □

**Organization:** This section will be organized as follows. In Subsection C.1, we prove a lemma quantifying the difference between the distributions of the regular and reflected Langevin Chains (Lemma 12). To use this lemma, we require a property of the Langevin diffusion for our setting (Lemma 13) — namely, that the chain with probability $1 - \epsilon$ remains in the region $\mathcal{D}$. We tackle this in Subsection C.2. In subsection C.3, we prove Lemma 4, namely that $p(z|x)$ restricted to $\mathcal{D}$ is log-concave. Finally, using the above three pieces, we will prove prove the main lemma of this section (Lemma 2) in subsection C.4.

We note that some of the techniques here have overlap with techniques used in (Moitra & Risteski, 2020), but we provide all the proofs in detail for completeness.

## C.1   RELATING RESTRICTED AND UNRESTRICTED LANGEVIN CHAINS

The main lemma of this section is as follows:

**Lemma 12** (Comparing with reflected Langevin diffusion). *Let $\tilde{z}_t$ follow the reflected Langevin SDE (Definition 4) with $g := L$), with stationary measure $\tilde{P}$ and with domain $\mathcal{D}$. Let $z_t$ follow the Langevin diffusion SDE (Definition 3) with $g := L$, with stationary measure $P$. Let $\tilde{P}_t$ be the density of $\tilde{z}_t$ and let $P_t$ be the density of $z_t$ and let $\tilde{P}_0$ be absolutely continuous with respect to the Lebesgue measure.*

*Then if $\forall t \geq 0, \mathbf{Pr}\left[\exists a \in [0,t], z_a \notin \mathcal{D}\right] \leq \alpha$, it holds that:*

$$d_{TV}(P_t, P) \lesssim 2\alpha + e^{-\frac{t}{2rad^2(\mathcal{D})}} \qquad \forall t \geq 0$$

*Proof.* Consider the total variation distance between $P_t$ and $P$. Using triangle inequality, we have

$$d_{\mathbf{TV}}(P_t, P) \leq d_{\mathbf{TV}}(P_t, \tilde{P}_t) + d_{\mathbf{TV}}(\tilde{P}_t, \tilde{P}) + d_{\mathbf{TV}}(\tilde{P}, P) \tag{21}$$

We first bound the second term using Theorem 3. Since $\mathcal{D}$ is convex and $L$ is convex over $\mathcal{D}$ by Lemma 4, we have

$$d_{\mathbf{TV}}(\tilde{P}_t, \tilde{P}) \lesssim e^{-\frac{t}{2rad^2(\mathcal{D})}} \tag{22}$$

For the first term, consider the coupling of $z_t$ and $\tilde{z}_t$ such that the Brownian motion $B_t$ is the same. We then have for any $t \geq 0$

$$d_{\mathbf{TV}}(P_t, \tilde{P}_t) \leq \mathbf{Pr}[z_t \neq \tilde{z}_t] \leq \mathbf{Pr}[\exists s \in [0,t], z_t \notin \mathcal{D}] \leq \alpha \tag{23}$$

where the first inequality follows by Lemma 6, and the second inequality follows from a rewrite that uses the fact that if the unrestricted chain $z_t$ stays inside the region $\mathcal{D}$, then the two diffusions won't differ at all due to the coupling of the Brownian motion. The term $\alpha$ bounds the probability of the unrestricted chain $z_t$ leaving the region $\mathcal{D}$.

For the third term, we need to bound $d_{\mathbf{TV}}(\tilde{P}, P)$. Note that $P : \mathbb{R}^d \to \mathbb{R}^+, \quad P(z) \propto e^{-L(z)}$ and $\tilde{P} : \mathcal{D} \to \mathbb{R}^+, \quad \tilde{P}(z) \propto e^{-L(z)}$. This means that $\forall z \in \mathcal{D}, \tilde{P}(z) > P(z)$ and $\forall z \in \mathbb{R}^d \setminus \mathcal{D}, P(z) > \tilde{P}(z)$. So we can write

$$d_{\mathbf{TV}}(\tilde{P}, P) = \mathbf{Pr}\left[z_\infty \notin \mathcal{D}\right] \tag{24}$$

where $z_\infty$ is the follows the stationary distribution of the Langevin diffusion $P$.

From the drift bound in the lemma statement, we can take the limit of $t \to \infty$ to get

$$\lim_{t \to \infty} \forall t \geq 0, \mathbf{Pr}\left[\exists a \in [0,t], z_a \notin \mathcal{D}\right] \leq \alpha$$
$$\implies \mathbf{Pr}\left[\exists a \in [0,\infty), z_a \notin \mathcal{D}\right] \leq \alpha$$
$$\implies \mathbf{Pr}\left[z_\infty \notin \mathcal{D}\right] \leq \alpha \tag{25}$$

Using equation equation 24 and equation 25, we get

$$d_{\mathbf{TV}}(\tilde{P}, P) \leq \alpha \tag{26}$$

The lemma statement follows from equations equation 23, equation 22 and equation 26. □

## C.2 Effective region for Langevin diffusion

In this section, we prove Langevin mostly stays in an effective region $\mathcal{D}$, namely:

**Lemma 13** (Effective region for Langevin diffusion). *Let* $\mathcal{D} := \{z : \|z - \hat{z}\| \leq rad(\mathcal{D})\}$ *with*

$$rad(\mathcal{D}) = \frac{4\beta}{m}\sqrt{\left(2d + \frac{\|x\|^2}{m^2}\right)\log\left(\frac{4\left(2d + \frac{\|x\|^2}{m^2}\right)}{\epsilon}\right)}. \; \text{Let } z_0 \text{ satisfy } \|z_0 - \hat{z}\| \leq \frac{1}{2}rad(\mathcal{D}). \text{ Then with}$$

$\beta \leq \beta_0$, *we have that* $\forall T > 0$, $\mathbf{Pr}\left[\exists t \in [0, T], \; z_t \notin \mathcal{D}\right] \leq \epsilon/4$, *where*

$$\beta_0 = \frac{4}{d\sqrt{\left(2d + \frac{\|x\|^2}{m^2}\right)\log\left(\frac{4\left(2d + \frac{\|x\|^2}{m^2}\right)}{\epsilon}\right)}} \cdot \min\left\{\frac{m^3}{6MM_2}, \frac{d^{0.75}m^2}{\sqrt{2MM_3}}\right\}$$

Towards proving Lemma 13, we will first prove Lemma 3.

*Proof of Lemma 3.* Using $dz = -\nabla L(z)dt + \sqrt{2}dB_t$ along with Itó's Lemma, we can compute

$$d\eta(z) = \langle \nabla\eta(z), -\nabla L(z)\rangle dt + \frac{1}{2} \cdot (\sqrt{2})^2 \Delta\eta(z)dt + \langle \nabla\eta(z), \sqrt{2}dB_t\rangle \tag{27}$$

We will bound each term on the right hand side. First note that $\nabla\eta(z) = z - \hat{z}$, and so $\|\nabla\eta(z)\| = \sqrt{2\eta(z)}$ for all $z \in \mathbb{R}^d$. We proceed to the second and third term since they're straightforward. For the third term, we have $\langle \nabla\eta(z), dB_t\rangle = \|\eta(z)\|dB_t = \sqrt{2\eta(z)}dB_t$, using a slight abuse of notation to denote the 1-dimensional Brownian motion (in the RHS) as also $B_t$. For the second term, note that $\Delta\eta(z) = d$ since $\nabla^2\eta(z) = I$. So:

$$d\eta(z) = -\langle \nabla\eta(z), \nabla L(z)\rangle dt + d \cdot dt + 2\sqrt{\eta(z)}dB_t \tag{28}$$

For the first term, from Lemma 26 we get

$$\langle \nabla\eta(z), \nabla L(z)\rangle = \langle z - \hat{z}, \nabla L(z)\rangle$$

$$= \langle z - \hat{z}, \hat{z}\rangle + \langle z - \hat{z}, \left(I + \frac{1}{\beta^2}J_G(\hat{z})^T J_G(\hat{z})\right)(z - \hat{z})\rangle + \langle z - \hat{z}, R_{\nabla L}(z)\rangle$$

$$= \langle z - \hat{z}, \hat{z}\rangle + \|z - \hat{z}\|^2 + \frac{1}{\beta^2}\|J_G(\hat{z})(z - \hat{z})\|^2 + \langle z - \hat{z}, R_{\nabla L}(z)\rangle \tag{29}$$

First, we analyze $\langle z - \hat{z}, \hat{z}\rangle$. Using Cauchy-Schwartz we can write $\langle z - \hat{z}, \hat{z}\rangle \geq -\|z - \hat{z}\|\|\hat{z}\|$.
Case 1. $\|z - \hat{z}\| \leq \|\hat{z}\|$. We can write that $\langle z - \hat{z}, \hat{z}\rangle \geq -\|\hat{z}\|^2$.
Case 2. $\|z - \hat{z}\| > \|\hat{z}\|$. We can write that $\langle z - \hat{z}, \hat{z}\rangle \geq -\|z - \hat{z}\|^2$.
In either case, it holds that

$$\langle z - \hat{z}, \hat{z}\rangle \geq -\left(\|z - \hat{z}\|^2 + \|\hat{z}\|^2\right) \tag{30}$$

Second, by Assumption 1, we can say that

$$\|J_G(\hat{z})(z - \hat{z})\|^2 \geq m^2\|z - \hat{z}\|^2 \tag{31}$$

Third, by the expression of the norm of the remainder from Lemma 26, we get

$$\langle z - \hat{z}, R_{\nabla L}(z)\rangle \geq -\|z - \hat{z}\| \cdot \|R_{\nabla L}(z)\|$$

$$\geq -\frac{1}{2\beta^2} \cdot \left(3dMM_2 \cdot \|z - \hat{z}\| + \sqrt{d}MM_3 \cdot \|z - \hat{z}\|^2\right) \cdot \|z - \hat{z}\|^2$$

For $z \in \mathcal{D}$ we have $\|z - \hat{z}\| \leq rad(\mathcal{D})$ so

$$\langle z - \hat{z}, R_{\nabla L}(z)\rangle \geq -\frac{1}{2\beta^2} \cdot \left(3dMM_2 \cdot rad(\mathcal{D}) + \sqrt{d}MM_3 \cdot rad(\mathcal{D})^2\right) \cdot \|z - \hat{z}\|^2 \qquad z \in \mathcal{D}$$

The inequality on $rad(\mathcal{D})$ from Lemma 11 implies that $3dMM_2 \cdot rad(\mathcal{D}) \leq \frac{m^2}{2}$ and $\sqrt{d}MM_3 \cdot rad(\mathcal{D})^2 \leq \frac{m^2}{2}$, and so

$$\langle z - \hat{z}, R_{\nabla L}(z)\rangle \geq -\frac{m^2}{2\beta^2}\|z - \hat{z}\|^2 \qquad z \in \mathcal{D} \tag{32}$$

Plugging in equation equation 30, equation 31, equation 32 in equation equation 29, we get

$$\langle \nabla \eta(z), \nabla L(z) \rangle \geq -\|\hat{z}\|^2 + \frac{m^2}{\beta^2}\|z - \hat{z}\|^2 - \frac{m^2}{2\beta^2}\|z - \hat{z}\|^2 = -\|\hat{z}\|^2 + \frac{m^2}{2\beta^2}\|z - \hat{z}\|^2 \qquad z \in \mathcal{D}$$

Using this in equation equation 28, and substituting the definition of $\eta(z)$, i.e. $\|z - \hat{z}\|^2 = 2\eta(z)$, finishes the proof. $\qquad\square$

Next, we prove high probability deviation bounds for Cox-Ingersoll-Ross processes. Precisely, we show:

**Lemma 14** (Concentration Bounds on Cox-Ingersoll-Ross Process)**.** *Consider the SDE*

$$dX_t = -wX_t dt + \tilde{N}dt + 2\sqrt{X_t}dB_t$$

*for $\tilde{N} \in \mathbb{N}$ and $w > 0$. Then for any $\epsilon > 0$*

$$\forall T > 0, \ \mathbf{Pr}\left[\forall t \in [0, T], \ s.t. \ X_t \leq 2X_0 + \frac{4\tilde{N}}{w}\log\left(\frac{4\tilde{N}}{\epsilon}\right)\right] \geq 1 - \epsilon$$

*Proof.* The SDE describes a Cox-Ingersoll-Ross process of dimension $\tilde{N}$, which equals in distribution to the random variable $Y_t$ as defined below, and with matching initial condition $Y_0 := X_0$.

$$Y_t = \sum_{i=1}^{\tilde{N}} \left(V_t^{(i)}\right)^2 \tag{33}$$

Here $V_t^{(i)}$ follow the Ornstein-Uhlenbeck equation $dV_t^{(i)} = -\frac{w}{2}V_t^{(i)}dt + dB_t^{(i)}$, and $V_0^{(i)} = \sqrt{Y_0/\tilde{N}}$ for all $i \in [\tilde{N}]$. $dB_t^{(i)}$ are (1-d) Brownian motions, and we have referenced them separately because they are independent for each $i \in [\tilde{N}]$. Indeed, applying Itó's Lemma, we recover the same SDE as:

$$dY_t = \sum_{i=1}^{\tilde{N}} \left[\left(2V_t^{(i)}\left(-\frac{w}{2}V_t^{(i)}\right) + \frac{1}{2} \cdot 2\right)dt + 2V_t^{(i)}dB_t^{(i)}\right]$$

$$= -w\sum_{i=1}^{\tilde{N}} \left(V_t^{(i)}\right)^2 dt + \tilde{N}dt + 2\sum_{i=1}^{\tilde{N}} V_t^{(i)}dB_t^{(i)}$$

$$= -wY_t dt + \tilde{N}dt + 2\sqrt{\sum_{i=1}^{\tilde{N}} \left(V_t^{(i)}\right)^2}dB_t$$

$$= -wY_t dt + \tilde{N}dt + 2\sqrt{Y_t}dB_t$$

Here in the RHS of the first expression, the term $\frac{1}{2} \cdot 2$ is the term $\frac{1}{2} \cdot \frac{\partial^2 \left(V_t^{(i)}\right)^2}{\partial B_t^{(i)2}}$ from Itó's Lemma. And in the third expression, notice that we have replaced a sum of brownian motions with a single one of matching variance.

Now we can write an explicit solution for the SDE. Since we know the solution of an Ornstein-Uhlenbeck process, each $V_t^{(i)}$ can be solved as (now we drop the indexing $i$ since the solution is same for each $i \in [\tilde{N}]$)

$$V_t = V_0 e^{-\frac{w}{2}t} + \underbrace{\int_0^t e^{-\frac{w}{2}(t-s)}dB_s}_{F(t)} \tag{34}$$

Applying the reflection principle to $F(t)$, we get

$$\forall T > 0, \forall a > 0 \qquad \mathbf{Pr}\left[\exists t \in [0, T], F(t) \geq a\right] = 2\mathbf{Pr}[F(T) \geq a] \tag{35}$$

Further, writing the distribution of $F(t)$, we have

$$F(t) \sim \mathcal{N}\left(0, \frac{1}{w}\left(1 - e^{-wt}\right)\right)$$

From standard Gaussian tail bounds, we have

$$\forall T > 0, \forall a > 0 \qquad \mathbf{Pr}\left[F(T) \geq a\right] \leq \exp\left(-\frac{a^2}{\frac{2}{w}\left(1 - e^{-wT}\right)}\right)$$

From equation 35, substituting $a := r\sqrt{\frac{2}{w}\left(1 - e^{-wT}\right)}$ we have

$$\forall T > 0, \forall r > 0 \qquad \mathbf{Pr}\left[\exists t \in [0, T], F(t) \geq r\sqrt{\frac{2}{w}\left(1 - e^{-wT}\right)}\right] \leq 2e^{-r^2}$$

Since the quantity $F(t)$ has a symmetric density about zero for all $t$, we can write the two sided bound as

$$\forall T > 0, \forall r > 0 \qquad \mathbf{Pr}\left[\exists t \in [0, T], |F(t)| \geq r\sqrt{\frac{2}{w}\left(1 - e^{-wT}\right)}\right] \leq 4e^{-r^2} \qquad (36)$$

Choosing $r$ such that $\epsilon/\tilde{N} = 4e^{-r^2}$ in equation 36, and using a union bound over the indices $i \in [\tilde{N}]$, we have

$$\forall T > 0 \quad \text{w.p. } (1-\epsilon): \ \forall t \in [0, T], \forall i \in [\tilde{N}], \ \left(V_t^{(i)} - V_0^{(i)} e^{-\frac{w}{2}t}\right)^2 \leq \log\left(\frac{4\tilde{N}}{\epsilon}\right) \cdot \frac{2}{w}\left(1 - e^{-wT}\right) \leq \frac{2}{w}\log\left(\frac{4\tilde{N}}{\epsilon}\right)$$

Plugging in $V_0^{(i)} = \sqrt{Y_0/\tilde{N}}$ for all $i \in [\tilde{N}]$, we get

$$\forall T > 0 \qquad \text{w.p. } (1 - \epsilon): \quad \forall t \in [0, T], \forall i \in [\tilde{N}], \ V_t^{(i)} \leq \sqrt{Y_0/\tilde{N}}\, e^{-\frac{w}{2}t} + \sqrt{\frac{2}{w}\log\left(\frac{4\tilde{N}}{\epsilon}\right)}$$

$$\leq \sqrt{Y_0/\tilde{N}} + \sqrt{\frac{2}{w}\log\left(\frac{4\tilde{N}}{\epsilon}\right)}$$

Finally, using $(a + b)^2 \leq 2(a^2 + b^2)$ and summing over all indices $i \in [\tilde{N}]$, we get

$$\forall T > 0 \qquad \text{w.p. } (1 - \epsilon): \quad \forall t \in [0, T], \ \sum_{i=1}^{\tilde{N}}\left(V_t^{(i)}\right)^2 \leq 2Y_0 + \frac{4\tilde{N}}{w}\log\left(\frac{4\tilde{N}}{\epsilon}\right) \qquad (37)$$

Replacing the sum by $Y_t$ using equation 33 and replacing $X_t, X_0$ in place of $Y_t, Y_0$ gives the result. $\qquad \square$

Next, since all of our SDEs so far have been actually inequalities, we will need a standard comparison theorem for SDEs:

**Lemma 15** (Comparison theorem, (Ikeda & Watanabe, 1977)). *Let $U_t$, $X_t$ be two stochastic processes following the SDEs*

$$dU_t = f(U_t)dt + h(U_t)dB_t$$

$$dX_t = g(X_t)dt + h(X_t)dB_t$$

*driven by the* same *Brownian motion $B_t$. Let at least one of these SDEs have a pathwise unique solution. If $\forall x \in \mathbb{R}^d$, $f(x) \leq g(x)$, and if $U_0 = X_0$, then with probability 1,*

$$U_t \leq X_t \qquad \forall t \geq 0$$

Lemma 15 shows domination of stochastic variables based on an inequality in their respective SDEs. This is almost what we need, since we showed an inequality in SDEs in Lemma 3 and showed that the upper bound is concentrated in Lemma 14. We need a slight adjustment on top of this though. Note that the inequality between SDEs in Lemma 3 holds only for $z \in \mathcal{D}$, however the inequality that the above Lemma requires is on all points $\alpha$, $f(\alpha) \leq g(\alpha)$. We can work through this by simply noting that conditioned on $X_t$ (the upper bound SDE) being concentrated in a region, $U_t$ will also be concentrated in the region. We state and show this below.

**Lemma 16.** *Let $U_t$, $X_t$ be two stochastic processes whose SDEs follow*

$$dU_t = f(U_t)dt + h(U_t)dB_t$$

$$dX_t = g(X_t)dt + h(X_t)dB_t$$

*driven by the* same *Brownian motion $B_t$. Let at least one of these SDEs have a pathwise unique solution. If $f(\alpha) \leq g(\alpha)$ for all $\alpha \leq \alpha_0$, and if $U_0 = X_0 \leq \alpha_0$; then*

$$\forall T > 0, \forall \bar{X} \leq \alpha_0 \quad \mathbf{Pr}\left[\forall t \in [0,T], U_t \leq \bar{X}\right] \geq \mathbf{Pr}\left[\forall t \in [0,T], X_t \leq \bar{X}\right]$$

*Proof.* Consider any $T > 0$ and $\bar{X} \leq \alpha_0$.

$$\mathbf{Pr}\left[\forall t \in [0,T], U_t \leq \bar{X}\right] \geq \mathbf{Pr}\left[\forall t \in [0,T], U_t \leq X_t \text{ and } \forall t \in [0,T], X_t \leq \bar{X}\right]$$

$$\geq \mathbf{Pr}\left[\forall t \in [0,T], X_t \leq \bar{X}\right] \cdot \underbrace{\mathbf{Pr}\left[\forall t \in [0,T], U_t \leq X_t \mid \forall t \in [0,T], X_t \leq \bar{X}\right]}_{1}$$

Where the last probability is one, because both $U_t$ and $X_t$ are continuous-time processes, and so $U_t > X_t$ given $X_t \leq \bar{X} \leq \alpha_0$ is not possible because $U_t$ cannot cross over $X_t$ in the $f \leq g$ region (using comparison Lemma 15). $\quad\square$

Given the above preliminaries, we can finally complete the proof of Lemma 13.

*Proof of Lemma 13.* Instantiate Lemma 16 with $U_t := \eta(z_t)$ and instantiate another SDE on $X_t$ with the same Brownian motion as $U_t$, as below

$$dX_t := -\frac{m^2}{\beta^2}X_t dt + \lceil \|\hat{z}\|^2 + d \rceil \cdot dt + 2\sqrt{X_t}dB_t$$

such that $X_0 := U_0 = \eta(z_0) = \frac{1}{2}\|z_0 - \hat{z}\|^2$. From Lemma 14 (plugging in $\tilde{N} := \lceil \|\hat{z}\|^2 + d \rceil$ and $w := \frac{m^2}{\beta^2}$), we can conclude that for any $\epsilon > 0$

$$\forall T > 0, \ \mathbf{Pr}\left[\forall t \in [0,T], \text{ s.t. } X_t \leq 2X_0 + \frac{4\tilde{N}}{w}\log\left(\frac{4\tilde{N}}{\epsilon}\right)\right] \geq 1 - \epsilon \tag{38}$$

Using the Lemma 16 on $U_t$ and $X_t$, with $\bar{X} := 2X_0 + \frac{4\tilde{N}}{w}\log\left(\frac{4\tilde{N}}{\epsilon}\right)$, and with $\alpha_0 := \frac{1}{2}rad(\mathcal{D})^2$ (since the inequality between SDEs of $U_t$ and $X_t$ holds in the region $U_t \leq \frac{1}{2}rad(\mathcal{D})^2$ from lemma 3), we get

$$\bar{X} \leq \alpha_0 \Rightarrow \forall T > 0, \ \mathbf{Pr}\left[\forall t \in [0,T], U_t \leq \alpha_0\right] \geq \mathbf{Pr}\left[\forall t \in [0,T], U_t \leq \bar{X}\right]$$

$$\geq \mathbf{Pr}\left[\forall t \in [0,T], X_t \leq \bar{X}\right]$$

$$\geq 1 - \epsilon$$

where the last inequality follows from equation 38. Note that $U_t \leq \alpha_0 \equiv \|z_t - \hat{z}\| \leq rad(\mathcal{D}) \equiv z_t \in \mathcal{D}$. This is exactly the event whose probability we need to lower bound.

We now check that the condition for the above is met, namely:

$$\bar{X} \leq \alpha_0 \equiv 2X_0 + \frac{4\tilde{N}}{w}\log\left(\frac{4\tilde{N}}{\epsilon}\right) \leq \frac{1}{2}rad(\mathcal{D})^2$$

Where $X_0 = \frac{1}{2}\|z_0 - \hat{z}\|^2 \le \frac{1}{2} \cdot \left(\frac{1}{2} rad(\mathcal{D})\right)^2$ from the lemma statement. This then becomes

$$\equiv \frac{4\tilde{N}}{w} \log\left(\frac{4\tilde{N}}{\epsilon}\right) \le \frac{1}{4} rad(\mathcal{D})^2, \text{ i.e.}$$

$$rad(\mathcal{D}) \ge 4\sqrt{\frac{\tilde{N}}{w} \log\left(\frac{4\tilde{N}}{\epsilon}\right)}$$

Putting in the values $\tilde{N}, w$ from above, and using Lemma 23 to bound $\|\hat{z}\|$ (since the RHS above is an increasing function of $\tilde{N}$), and bounding the ceiling trivially by $\lceil \kappa \rceil \le \kappa + d$ (since $d \ge 1$) means it suffices that

$$rad(\mathcal{D}) \ge \frac{4\beta}{m}\sqrt{\left(2d + \frac{\|x\|^2}{m^2}\right) \log\left(\frac{4\left(2d + \frac{\|x\|^2}{m^2}\right)}{\epsilon}\right)}$$

Using the bound on $\beta$ the above inequality is indeed satisfied for our choice of $rad(\mathcal{D})$. $\qquad\square$

## C.3 PROOF OF LEMMA 4

*Proof.* Using the expression of $\nabla^2 L$ from equation equation 56, we get that

$$\nabla^2 L(z) \succeq \left(1 + \frac{1}{\beta^2}\left(m^2 - \left\|\sum_{i\in[d]} \nabla^2 G_i(z)\left(G_i(z) - x_i\right)\right\|_{op}\right)\right) I \tag{39}$$

And we have

$$\left\|\sum_{i\in[d]} \nabla^2 G_i(z)\left(G_i(z) - x_i\right)\right\|_{op} \le \sum_{i\in[d]} |G_i(z) - x_i| \cdot \left\|\nabla^2 G_i(z)\right\|_{op}$$

$$\le \sum_{i\in[d]} |G_i(z) - x_i| \cdot \left\|\nabla^2 G_i(z)\right\|_{op}$$

$$\le^{(0)} \sqrt{d} \cdot \|G(z) - x\| \cdot \left\|\nabla^2 G(z)\right\|_{op}$$

$$\le^{(1)} \sqrt{d} \cdot M\|z - \hat{z}\| \cdot M_2$$

$$\le \sqrt{d}MM_2 \cdot rad(\mathcal{D}) \qquad z \in \mathcal{D}$$

In $(0)$, we used the fact that $\forall i \in [d], \left\|\nabla^2 G_i(z)\right\|_{op} \le \|\nabla^2 G(z)\|_{op}$; which can be derived from the sup-definition of the tensor $\ell_2$ norm from (Lim, 2005). In $(1)$, we've used Assumption 2. Using the expression of $rad(\mathcal{D})$ from Lemma 11, we get

$$\left\|\sum_{i\in[d]} \nabla^2 G_i(z)\left(G_i(z) - x_i\right)\right\|_{op} \le \frac{m^2}{6\sqrt{d}} \qquad z \in \mathcal{D}$$

$$\le m^2 \qquad z \in \mathcal{D} \ \text{ (since } d \ge 1\text{)}$$

Using this in equation 39, we get $\nabla^2 L(z) \succeq I$ for $z \in \mathcal{D}$. $\qquad\square$

## C.4 PROOF OF LEMMA 2

We can now put together the proof of Lemma 2.

*Proof of Lemma 2.* Using Lemma 12 with $\mathcal{D}$ defined as in Lemma 13, with $\alpha := \epsilon/4$ in Lemma 13 and setting $T = rad^2(\mathcal{D}) \log \frac{1}{\epsilon}$ we get $e^{-T/2rad^2(\mathcal{D})} \le \frac{\epsilon}{2}$. Using $rad(\mathcal{D}) = \mathcal{O}(1/d)$ from lemma 11, the result follows. $\qquad\square$

# D   PROOF OF THEOREM 1: EULER-MARIYAMA DISCRETIZATION AND NEURAL SIMULATION

In this section, we use the mixing time bounds in the previous sections, along with a discretization error analysis when running a disretized (in the simplest manner, via a Euler-Mariyama scheme) version of the Langevin diffusion chain. This is done in Subsection D.1. Subsequently, we show that each of these steps can be simulated by a "layer" of a neural network. This is done in Subsection D.2. Finally, we put everything together to prove Theorem 1 in Subsection D.3.

## D.1   BOUNDING THE ERROR IN THE EULER-MARIYAMA DISCRETIZATION

We finally wish to show that with a polynomially small discretization level (i.e. $h$), the Euler-Mariyama scheme approximates appropriately for our purposes the Langevin diffusion process. We recall from equation 2 that this Markov process is described by

$$z \leftarrow z - h\nabla L(z) + \sqrt{2h}\xi_t, \ \ \xi \sim N(0, I) \tag{40}$$

There is a rich field in the literature on discretizing Langevin diffusion (Dalalyan, 2016; Raginsky et al., 2017; Dalalyan, 2017) primarily used for designing efficient sampling algorithms — a large fraction of them analyzing the above discretization. We will mostly rely on tools from Bubeck et al. (2018). We show:

**Lemma 17** (Euler-Mariyama discretization). *Let $z_t$ follow the Langevin diffusion $dz_t = -\nabla L(z_t)dt + \sqrt{2}dB_t$, and let $P_t$ denote the distribution of $z_t$.*

*Let $\hat{z}_t$ follow the continuous-time Markov process determined by the Euler-Mariyama discretization scheme with discretization level $h$ given by equation 12, namely*

$$d\hat{z}_t = -\nabla L(\hat{z}_{\lfloor t/h \rfloor h})dt + \sqrt{2}dB_t \tag{41}$$

*Let us denote by $\hat{P}_t$ the distribution of $\hat{z}_t$. Finally, let $z_0 = \hat{z}_0 \in \mathcal{D}$ satisfying $\|z_0 - \hat{z}\| \leq \frac{1}{2}rad(\mathcal{D})$, with $\mathcal{D}$ defined as in Lemma 13.*

*Then, if $T = (2rad(\mathcal{D}))^2 \log(1/\epsilon)$ and $h = poly(\epsilon, 1/d, 1/M, m, \beta)$, we have $d_{TV}(\hat{P}_T, P_T) \leq \epsilon$*

*Proof.* Let $y_t$ follow the Reflected Langevin SDE (Definition 4), with $g := L$ and region $\mathcal{D}$, and with the initialization $y_0 = z_0$. Let $\hat{y}_t$ follow the continuous-time extension of the discretized reflected Langevin SDE given by equation 14 with $g := L$ and region $\mathcal{D}$), with the initial point $\hat{y}_0 = z_0$. Let $Q_t$ and $\hat{Q}_t$ denote the respective distributions. Let $Q$ denote the stationary distribution of $y_t$.

From triangle inequality, we can decompose our error as

$$d_{\mathbf{TV}}\left(\hat{P}_T, P_T\right) \leq d_{\mathbf{TV}}\left(Q_T, P_T\right) + d_{\mathbf{TV}}\left(\hat{Q}_T, Q_T\right) + d_{\mathbf{TV}}\left(\hat{P}_T, \hat{Q}_T\right) \tag{42}$$

We will bound each of the terms in turn.

For the first term, consider a coupling of $y_t$ and $z_t$ that uses the same Brownian motion for both processes, so long as neither has left $\mathcal{D}$, and chooses an arbitrary coupling henceforth. By Lemma 6, we have

$$
\begin{aligned}
d_{\mathbf{TV}}(Q_T, P_T) &\leq \mathbf{Pr}[z_T \neq y_T] \\
&\leq \mathbf{Pr}[z_T \text{ leaves } \mathcal{D}] + \mathbf{Pr}[y_T \text{ leaves } \mathcal{D}] \\
&\leq \epsilon/2
\end{aligned}
$$

where the last inequality follows by Lemma 13.

For the second term, we can further break it up, again by using the triangle inequality as

$$d_{\mathbf{TV}}\left(\hat{Q}_T, Q_T\right) \leq d_{\mathbf{TV}}\left(\hat{Q}_T, Q\right) + d_{\mathbf{TV}}\left(Q, Q_T\right) \tag{43}$$

We will use a result about the discretization of reflected Langevin diffusion from (Bubeck et al., 2018). Towards that, let us denote

$$L_{\mathcal{D}}^1 = \max_{z \in \mathcal{D}} \|\nabla L(z)\| \tag{44}$$

$$L_{\mathcal{D}}^2 = \max_{z \in \mathcal{D}} \|\nabla^2 L(z)\|_{op} \tag{45}$$

Using Theorem 1 from (Bubeck et al., 2018), if $T = (2\text{rad}(\mathcal{D}))^2 \log(1/\epsilon)$ and $h = \text{poly}(\epsilon, 1/d, 1/L_{\mathcal{D}}^1, 1/L_{\mathcal{D}}^2)$, both terms in equation 43 are upper bounded by $\epsilon/4$. On the other hand, by equations equation 55 and equation 56, we have

$$\|\nabla L(z)\| \leq \|z\| + \frac{1}{\beta^2}\|J_G(z)^T\|_2\|G(z) - x\|$$

$$\leq (\|\hat{z}\| + rad(\mathcal{D})) + \frac{1}{\beta^2}M^2 rad(\mathcal{D})^2 \qquad z \in \mathcal{D}$$

Using Lemma 23 and the first term of the RHS from Lemma 11, we get

$$\|\nabla L(z)\| \leq \frac{\|x\|}{m} + \left(1 + \frac{M^2}{\beta^2}\right)\frac{m^4}{36d^2 M^2 M_2^2} \qquad z \in \mathcal{D}$$

$$\leq \mathcal{O}(\sqrt{d}) + \mathcal{O}\left(\frac{1}{\beta^2 d^2}\right) \qquad z \in \mathcal{D}$$

Which follows using Lemma 24. For the second order bound, a similar calculation as the proof of Lemma 4 we get

$$\|\nabla^2 L(z)\|_{op} \leq 1 + \frac{1}{\beta^2}\left(\|J_G(z)\|_{op}^2 + \sum_i \|\nabla^2 G_i\|_2|G_i(z) - x_i|\right)$$

$$\leq 1 + \frac{1}{\beta^2}(M^2 + m^2)$$

$$= \mathcal{O}\left(\frac{1}{\beta^2}\right)$$

Plugging the bounds above, we get that if $T = (2\text{rad}(\mathcal{D}))^2 \log(1/\epsilon)$ and $h = \text{poly}(\epsilon, 1/d, 1/M, m, \beta)$, then $d_{\textbf{TV}}\left(\hat{Q}_T, Q_T\right) \leq \epsilon/2$.

Finally, for the third term, we will show that

$$d_{\textbf{TV}}\left(\hat{P}_T, \hat{Q}_T\right) \leq \epsilon/4 \tag{46}$$

Towards that, by Lemma 6, coupling the Gaussian noise to be the same for $\{\hat{P}_t\}_{t \in [T]}$ and $\{\hat{Q}_t\}_{t \in [T]}$, we have

$$d_{\textbf{TV}}\left(\hat{P}_T, \hat{Q}_T\right) \leq \Pr[\hat{P}_T \neq \hat{Q}_T]$$

$$\leq \Pr[\exists t \in [T], \hat{z}_t \notin \mathcal{D}]$$

By the definition of total variation distance, and Lemma 6 again, we have

$$\Pr[\exists t \in [T], \hat{z}_t \notin \mathcal{D}] \leq d_{\textbf{TV}}\left(\{\hat{z}_t\}_{t \in [T]}, \{z_t\}_{t \in [T]}\right) + \Pr[\exists t \in [T], z_t \notin \mathcal{D}]$$

By Lemma 13 and Lemma 6, we have $\Pr[\exists t \in [T], z_t \notin \mathcal{D}] \leq \epsilon/4$ and $d_{\textbf{TV}}\left(\{\hat{z}_t\}_{t \in [T]}, \{z_t\}_{t \in [T]}\right) \leq \epsilon/4$ which proves equation 46.

$\square$

## D.2 NEURAL SIMULATION OF ARITHMETIC OPERATIONS

The deep latent Gaussian will be an overall map from observables $x \in \mathbb{R}^d$ to latents $z \in \mathbb{R}^d$, such that $z$ is an approximate sample from the posterior $p(z|x)$.

In this section, we prove "simulation" lemmas that show how to combine neural networks for two functions $f, g$ to create a neural network for some arithmetic operation applied to $f, g$ (e.g. $f + g$, $f \times g$, etc.).

Recall, $\rho : \mathbb{R} \to \mathbb{R}$ denotes the square activation function, i.e. $\rho(x) = x^2$.

**Lemma 18.** *If $f : \mathbb{R}^d \to \mathbb{R}$ is a neural network with $M$ parameters and $g : \mathbb{R}^d \to \mathbb{R}$ is a neural network with $N$ parameters, both with set of activation functions $\Sigma$, then $w_1 f + w_2 g$ can be represented by a neural network with $\mathcal{O}(M + N)$ parameters and set of activation functions $\Sigma$ for any $w_1, w_2 \in \mathbb{R}$.*

*Proof.* We simply note that we can make a network with one more layer than the maximum of the depth of $f$ and $g$ that adds up the scaled outputs of $f$, $g$ to produce $w_1 f + w_2 g$. No additional nonlinearity is needed. $\qquad\square$

**Lemma 19.** *If $f : \mathbb{R}^d \to \mathbb{R}$ is a neural network with $M$ parameters and $g : \mathbb{R}^d \to \mathbb{R}$ is a neural network with $N$ parameters, both with set of activation functions $\Sigma$, then for any $w \in \mathbb{R}$, $v = wfg$ can be represented by a neural network with $\mathcal{O}(M + N)$ parameters with set of activations $\Sigma \cup \{\rho\}$.*

*Proof.* Noting $xy = \frac{1}{4}\left((x + y)^2 - (x - y)^2\right)$ for all $x, y \in \mathbb{R}$, we can implement the multiplication of scalars using a two-layer neural network with $\mathcal{O}(1)$ parameters using the $\rho$ activation function. We then proceed same as for addition, adding one layer to multiply the outputs of $f$, $g$. $\qquad\square$

**Lemma 20.** *If $G : \mathbb{R}^d \to \mathbb{R}^d$ is represented by a neural network with $N$ parameters and differentiable activation function set $\Sigma$, given an $x \in \mathbb{R}^d$ and $z \in \mathbb{R}^d$, one can compute for any $c_1, c_2 \in \mathbb{R}$:*

$$c_1 z + c_2 J_G(z)^T \left(G(z) - x\right)$$

*via a neural network with $\mathcal{O}(Nd^2)$ parameters and activation functions $\Sigma \cup \Sigma' \cup \{\rho\}$, where $\Sigma'$ contains the derivatives of the activations in $\Sigma$.*

*Proof.* Each coordinate of $J_G : \mathbb{R}^d \to \mathbb{R}^{d \times d}$ can be written as a neural network with $\mathcal{O}(N)$ parameters by Lemma 5, with activation functions $\Sigma \cup \Sigma'$. Each coordinate of $G(z) - x$ can be computed in $\mathcal{O}(N)$ parameters using lemma 18.

Performing the multiplications needed for $J_G(z)^T \left(G(z) - x\right)$ via Lemma 19 results in a network of size $\mathcal{O}(Nd^2)$ due to the $d^2$ size of the Jacobian, and together with the addition of $c_1 z$ (via $\mathcal{O}(d)$ parameters) results in the final bound of $\mathcal{O}(Nd^2)$ on the parameters. $\qquad\square$

### D.3 PROOF OF THEOREM 1

Finally, we can put all the ingredients together to prove Theorem 1.

*Proof of Theorem 1.* The deep latent Gaussian we construct will have all latent dimensions $d$ as well as the output dimension $d$. The input will be $x \in \mathbb{R}^d$, the observable on which we want to carry out the inference. We will first create a part for the GD simulation, and then one for the Langevin chain.

Using Lemma 1 with $\delta := \frac{1}{4} rad(\mathcal{D})$, if we run Gradient Descent for $S = \mathcal{O}(\frac{d^2}{rad(\mathcal{D})^2}) = \mathcal{O}(\frac{d}{\beta^2})$ steps (using the bound $rad(\mathcal{D}) = \Omega(\beta\sqrt{d})$ from Lemma 13), we can get a $z_{init}$ with $\|z_{init} - \hat{z}\| \leq \frac{1}{4} rad(\mathcal{D})$.

Using Lemma 20, we can construct a network $D_{one}$ with $\mathcal{O}\left(Nd^2\right)$ parameters that executes a single step of gradient descent. In the deep latent Gaussian we construct, we repeat $D_{one}$ with zero variance $\mathcal{O}(\frac{d}{\beta^2})$ times to simulate all the steps of gradient descent and construct $z_{init}$. The overall number of parameters of the deep latent Gaussian so far is then $\mathcal{O}\left(\text{poly}(d, \frac{1}{\beta}) \cdot N\right)$. Note, though the first variable $z_0$ has unit variance, we can easily transform it to a zero variance Gaussian to simulate the first step by multiplying by 0.

Starting with the random point $z_0$, if we run the discretized Langevin dynamics given by equation 11 for $T/h$ steps with $T$ and $h$ as given by Lemma 17, we get by the error bounds of Lemma 17 and Lemma 2 and the triangle inequality:

$$d_{\text{TV}}\left(\hat{P}_T, p(z \mid x)\right) \leq \epsilon$$

which is the required overall bound.

We further note that each step of the discretized Langevin dynamics can be simulated by a layer of the deep latent Gaussian: by Lemma 20 we can construct a network $S_{one}$ with $\mathcal{O}\left(Nd^2\right)$ parameters that computes the next Langevin iterate. Choosing the variance to be $2h$, then, accomplishes exactly the distribution of a single step of the discretized dynamics. Given the choice of $T$ and $h$, the number of steps we need to run is $\mathcal{O}\left(\text{poly}(1/\epsilon, d, 1/\beta)\right)$. Hence, the overall number of parameters required for simulating the discretized dynamics is $\mathcal{O}\left(\text{poly}(1/\epsilon, d, 1/\beta) \cdot N\right)$. Finally, it's clear that the dependence on $x$ in the layers comes through the $J_G(z)^T(G(z) - x)$ terms in the gradient and Langevin updates—so the weights for the deep latent Gaussian can be produced by a network of size $\mathcal{O}\left(\text{poly}(1/\epsilon, d, 1/\beta) \cdot N\right)$ as well. The multiplication by $x$ can be implemented via the square activation $\rho$. Thus, the proof is finished $\qquad\square$

## E    Proof of Theorem 2: Lower bound

First, we observe the following proposition, which shows that the samples $x$ with high probability lie near a vertex of the hypercube $\{\pm 1\}^d$

**Lemma 21** (Closeness to hypercube). *A sample $x$ from the VAE with $G(z) = \mathcal{C}(sgn(z))$ and variance $\beta^2 I$ satisfies, for all $c > 1$*

$$\mathbf{Pr}[\|x - G(z)\| \leq 6c\beta\sqrt{d}] \geq 1 - \exp(-c^2 d)$$

*Proof.* Since $x - G(z)$ is a Gaussian with variance $\beta^2 I$, the claim follows via standard concentration of the norm of a Gaussian. $\qquad\square$

Furthermore, we also have the following lemma, that shows that when $x$ is near a vertex of the hypercube, the posterior $p(z|x)$ concentrates near the $\mathcal{C}^{-1}(sgn(x))$, that is, the pre-image of the nearest point to $x$ on the hypercube.

**Lemma 22** (Concentration near pre-image). *Let $x$ be such that $\|x - sgn(x)\| \leq 6\beta\sqrt{d}$. Then,*

$$\mathbf{Pr}_{z\sim p(z|x)}[sgn(z) \neq \mathcal{C}^{-1}(sgn(x))] \leq exp(-2d)$$

*Proof.* By Bayes rule, since $p(sgn(z))$ is uniform, we have

$$p(sgn(z)|x) \propto p(x|sgn(z))$$

Consider then any $b \in \{\pm 1\}^d, b \neq sgn(z)$. We have:

$$\frac{p(x|b)}{p(x|sgn(z))} = \frac{e^{-1/\beta^2 \|x-b\|^2}}{e^{1/\beta^2\|x-sgn(z)\|^2}}$$
$$= e^{-1/\beta^2\left(\|x-b\|^2 - \|x-sgn(z)\|^2\right)}$$
$$\leq e^{-10d}$$

where the last inequality follows for $\beta = o(1/\sqrt{d})$ and a small enough constant in the $o(\cdot)$ notation. Hence,

$$p(sgn(z)|x) = 1 - \sum_{b \neq sgn(z)} p(b|x)$$
$$\geq 1 - 2^d e^{-10d} p(x|sgn(z))$$
$$\geq 1 - e^{-2d}$$

$\qquad\square$

*Proof of Theorem 2.* The proof will proceed by a reduction to Conjecture 1. Namely, $\mathcal{C}$ be a Boolean circuit satisfying the one-way-function assumptions in Conjecture 1. Let $G$ be defined as $G(z) = \mathcal{C}(sgn(z))$

Suppose that there is an encoder $E$ of size $T(n)$, s.t. with probability at least $\epsilon(d)$ over the choice of $x$ (wrt to the distribution of $G$), we have $\text{TV}(p(z|x), q(z|x)) \leq 1/10$, where $q(z|x)$ is the distribution of the encoder when $x$ is supplied as input.

We will first show how to construct a neural sequential Gaussian $\mathcal{C}''$ which violates Conjecture 1 in an "average sense": namely, for $1/\text{poly}(d)$ fraction of inputs $\tilde{x} \in \{\pm 1\}^d$, with probability $1 - \exp(-d)$ it outputs the $\tilde{z}$, s.t. $\tilde{x} = \mathcal{C}(\tilde{z})$.

Subsequently we will remove the randomness — i.e. we will produce from this a deterministic and a neural network, rather than a circuit – subsequently we will show how to remove both the randomness, and use simulate the neural networks using a Boolean circuit.

This deep latent Gaussian $\mathcal{C}''$ receives a $\tilde{x} \in \{\pm 1\}^d$ and constructs (randomly) a multiple samples $x \in \mathbb{R}^d$ as

$$x = \tilde{x} + \xi, \quad \xi \sim N(0, \beta^2 I_d) \tag{47}$$

Subsequently, it samples (multiple) $z$ from $E(z|x)$ and checks whether $\tilde{x} = \mathcal{C}(\text{sgn}(z))$. If yes, it outputs $\text{sgn}(z)$. If for none of the samples $x$ and $z$, $\text{sgn}(z) = \tilde{x}$, it outputs $(1, 1, \ldots, 1)$.

We will show that for $1/\text{poly}(d)$ fraction of inputs $\tilde{x} \in \{\pm 1\}^d$, with probability $1 - \exp(-d)$, $\tilde{\mathcal{C}}''$ outputs the $\tilde{z}$, s.t. $\tilde{x} = \mathcal{C}(\tilde{z})$.

Note that $x$ is distributed according to the distribution of the generator $G$. (Since $\tilde{x}$ is uniformly distributed on the hypercube.) Let $E_1$ be the event that $d_{\text{TV}}(p(z|x), E(z)) \leq \frac{1}{10}$, $E_2$ the event that $\|x - \text{sgn}(x)\| \leq 6\beta\sqrt{d}$. By Assumption, $\mathbf{Pr}[\bar{E}_1] \leq 1 - \epsilon(d)$; by Lemma 21, for a large enough $d$, $\mathbf{Pr}[\bar{E}_2] \leq \frac{\epsilon(d)}{2}$. Hence, $\mathbf{Pr}[\bar{E}_1 \vee \bar{E}_2] \leq 1 - \frac{\epsilon(d)}{2}$., i.e. $\mathbf{Pr}[E_1 \wedge E_2] \geq \frac{\epsilon(d)}{2}$.

Consider an $x$ for which both events $E_1$ and $E_2$ attain. From the the definition of TV distance and Lemma 22, we have

$$\mathbf{Pr}_{z \sim E(z|x)}[\text{sgn}(z) \neq \mathcal{C}^{-1}(\text{sgn}(x))] \leq \mathbf{Pr}_{z \sim p(z|x)}[\text{sgn}(z) \neq \mathcal{C}^{-1}(\text{sgn}(x))] + \frac{1}{10}$$

$$\leq \exp(-2d) + \frac{1}{10}$$

$$\leq \frac{1}{5}$$

Hence, we have

$$\mathbf{Pr}_x\left[\mathbf{1}\left(\mathbf{Pr}_{z \sim E(z|x)}[\text{sgn}(z) \neq \mathcal{C}^{-1}(\text{sgn}(x))] \leq 1/5\right)\right] \geq \frac{\epsilon(d)}{2}$$

By the tower rule of probability, we can rewrite this as

$$\mathbb{E}_{\hat{z} \sim N(0, I_d)}\left[\mathbf{Pr}_{x \sim p(x|\hat{z})}\left[\mathbf{1}\left(\mathbf{Pr}_{z \sim E(z|x)}[\text{sgn}(z) \neq \mathcal{C}^{-1}(\text{sgn}(x))] \leq 1/5\right)\right]\right] \geq \epsilon(d)/2$$

Let us denote the random variable

$$\mathcal{A}(\hat{z}) := \mathbf{Pr}_{x \sim p(x|\hat{z})}\left[\mathbf{1}\left(\mathbf{Pr}_{z \sim E(z|x)}[\text{sgn}(z) \neq \mathcal{C}^{-1}(\text{sgn}(x))] \leq 1/5\right)\right]$$

such that we have

$$\mathbb{E}_{\hat{z}}[\mathcal{A}(\hat{z})] \geq \epsilon(d)/2$$

By the reverse Markov inequality, it follows that

$$\mathbf{Pr}[\mathcal{A}(z) \leq a] \leq \frac{1 - \epsilon(d)/2}{1 - a}$$

Taking $a = \epsilon(d)/2$ we get $\mathbf{Pr}[\mathcal{A}(z) \leq \epsilon(d)/2] \leq 1 - \epsilon^2(d)/4$. Hence, with probability at least $\epsilon^2(d)/4$ over the choice of $\hat{z}$, we have

$$\mathbf{Pr}_{x \sim p(x|\hat{z})}\left[\mathbf{1}\left(\mathbf{Pr}_{z \sim E(z|x)}[\text{sgn}(z) \neq \mathcal{C}^{-1}(\text{sgn}(x))] \leq 1/5\right)\right] \geq \epsilon(d)/2$$

As we indicated, $\mathcal{C}''$ will repeat sampling $x$ and $z|x$ multiple times. Let us denote by $\xi_1$ the Gaussian noise sampled in equation 47 and $\{\xi_2^i\}_{i=1}^L$ the Gaussian samples used in sampling $z \sim q(z|x)$. Consider sampling $\{\xi_1^i, i \in [1, M]\}$, where $M = \Omega\left(\frac{1}{\epsilon(d) \log d}\right)$ and $\{\xi_2^{i', j}, i' \in [L], j \in [M']\}$,

where $M' = \Omega(\log d)$, and for each $i \in [M], j \in [M']$ outputs $x = \hat{x} + \xi_1^i$, and samples from $q(z|x)$ using the randomness in $\{\xi_2^{i',j}\}_{i'=1}^L$. By Chernoff bounds, with probability $1 - \exp(-\Omega(d))$, at least one pair of indices $(i, j)$ is such that the corresponding choice of random vectors results in a choice of $z$, such that $\mathcal{C}(\mathrm{sgn}(z)) = \hat{x}$. Since the total number of possible strings $\hat{x}$ is $2^d$, taking the constants in $M, M'$ sufficiently large, by union bound, there is a pair $(i, j)$, s.t. the corresponding vectors $\xi_1^i, \{\xi_2^{i',j}\}_{i'=1}^L$ works for all $\hat{x}$. Hard-coding these into the weights of a neural network, we get a *deterministic* neural network of size $O(|\mathcal{C}| + T(d))$, that with probability $\epsilon^2(d)/4$ inverts $\mathcal{C}$.

The only leftover issue is that $E$ is a neural network (hence, a continuous function), whereas we are trying to produce a Boolean circuit. Conversions between neural networks applied to Boolean inputs and Boolean circuits are fairly standard (see Siegelmann (2012), Chapter 4, Theorem 6), but we include a proof sketch for completeness nevertheless.

First, we show that having logarithmic in $L, W, M$ precision for the weights and activations in $E$ suffices. For notational convenience, let us denote by $N$ the maximum degree of any node in $E$. The proof is essentially the same as Lemma 4.2.1 in Siegelmann (2012) – the only difference is that our inputs are close to binary, but not exactly binary. Namely, for some constants $\delta_w, \delta_a$ to be chosen, consider a network $\hat{E} : \mathbb{R}^d \to \mathbb{R}^d$, which has the same architecture as $E$, and additionally:

- The edge weights $\{\tilde{w}_{i,j}\}$ are produced by truncating the weights $w_{i,j}$ to $\log(\delta_w)$ significant bits, s.t. $\forall i, j : |w_{i,j} - \tilde{w}_{i,j}| \leq \delta_w$.

- If a node in $E$ has activation $\sigma$, the corresponding node in $E'$ has activation $\sigma'$ that truncates the value of $\sigma$ to $\log(\delta_a)$ significant bits, s.t. $\forall x, |\sigma(x) - \sigma'(x)| \leq \delta_a$.

Then, we claim that for nodes at depth $t$ from the input, denoting $v(x)$ and $\hat{v}(x)$ the function calculated by the node in $E$ and $\hat{E}$ respectively, we will show by induction:
$$|v(x) - \hat{v}(x)| \leq L(LNW)^{t-1}(LNM\delta_w + \delta_a)$$
Let us denote by $\epsilon_t$ the maximum of $|v(x) - \hat{v}(x)|$ for any nodes $v, \hat{v}$ at depth $t$. Suppose the claim is true for nodes at depth $t$ and consider a node at depth $t + 1$. Denoting by $v_1, v_2, \ldots, v_n$ the inputs to $v$, we have

$$|v(x) - \hat{v}(x)| = \left| \sigma\left(\sum_i w_i v_i(x)\right) - \sigma'\left(\sum_i \hat{w}_i \hat{v}_i(x)\right) \right| \tag{48}$$

$$\leq \left| \sigma\left(\sum_i w_i v_i(x)\right) - \sigma\left(\sum_i \hat{w}_i \hat{v}_i(x)\right) \right| + \left| \sigma'\left(\sum_i \hat{w}_i \hat{v}_i(x)\right) - \sigma'\left(\sum_i \hat{w}_i \hat{v}_i(x)\right) \right| \tag{49}$$

$$\leq L \sum_i |w_i v_i(x) - \hat{w}_i \hat{v}_i(x)| + \delta_a \tag{50}$$

$$\leq L \sum_i (|w_i v_i(x) - w_i \hat{v}_i(x)| + |\hat{w}_i \hat{v}_i(x) - w_i \hat{v}_i(x)|) + \delta_a \tag{51}$$

$$\leq L \sum_i (W\epsilon_t + M\delta_w) + \delta_a \tag{52}$$

$$\leq LNW\epsilon_t + LNM\delta_w + \delta_a \tag{53}$$

where equation 50 follows by Lipschitzness of $\sigma$ and equation 52 follows from the fact that the weights are bounded by $W$ and the outputs $v, \hat{v}$ are bounded by $M$.

Unfolding the recursion, we get

$$\epsilon_{t+1} \leq \sum_{i=0}^t (LNW)^i (LNM\delta_w + \delta_a) \leq (LNW)^t (LNM\delta_w + \delta_a)$$

Given this estimate, we can choose $\delta_w, \delta_a$, s.t. the values at each output coordinate of $E$ and $\hat{E}$ match. It suffices to have $(LNW)^d(LNM\delta_w + \delta_a) < 1$, which obtains when $\delta_w, \delta_a = O\left(\frac{1}{M}(LNW)^{-d}\right)$ – i.e. it suffices to only keep $O\left(d \log(LNW) + \log(M)\right)$ significant bits.

Next, we show how to simulate $\hat{E}$ by a Boolean circuit – this is essentially part of the proof of Lemma 4.2.2 in Siegelmann (2012). Namely, each pre-activation can be computed by a subcircuit of size $O\left(N\left(d\log(LNW) + \log(M)\right)^2\right)$ by the classic carry-lookahead algorithm for addition. Since we only require $\log(\delta_a)$ bits of accuracy for the activation function, and each activation is at most $M$ (so we need $O(\log M)$ bits for the integer part), we need $O(\log M + \log(\delta_a))$ nodes for the activation function.

Hence, we can simulate $\hat{E}$ by a Boolean circuit which is at most a factor of poly $(N, \log(L), \log(W), \log(M))$. Since $L, W, M$ are $o(\exp(\mathrm{poly}(d)))$, the size of the size of the resulting Boolean circuit is still poly$(d)$. Thus, we've constructed a polynomial-sized circuit which inverts the one-way-permutation, thus violating Conjecture 1.

$\square$

## F    TECHNICAL LEMMAS

This section contains several equations and calculational lemmas that we repeatedly use across the proofs of Theorem 1.

### F.1    BOUND ON $\|\hat{z}\|$

Recall that we defined the inverse point $\hat{z}$ for a given $x \in \mathbb{R}^d$ in equation equation 4. As mentioned earlier, due to bijectivity, it is unique and satisfied $G(\hat{z}) = x$. Further, we can easily prove a bound on the norm of $\hat{z}$ using our assumptions, which will be useful in various places in the proof.

**Lemma 23.** *$\hat{z}$ defined as in equation 4 with $G$ that satisfies Assumption 1, is such that*

$$\|\hat{z}\| \leq \frac{\|x\|}{m}$$

*Proof.* Using strong invertibility from Assumption 1 and centering from Assumption 3, we can write

$$\|x - G(0)\| = \|G(\hat{z}) - G(0)\| \geq m \cdot \|\hat{z} - 0\|$$
$$\implies \|\hat{z}\| \leq \frac{\|x\|}{m} \tag{54}$$

$\square$

### F.2    HIGH PROBABILITY BOUND ON $\|x\|$

**Lemma 24** (High-probability bound on $\|x\|$)**.** *With $x$ following the density from Definition 1 (with $d_l = d_o = d$), s.t. $G$ satisfies Assumption 2, we have that*

$$\|x\| \leq 12(M + \beta)\sqrt{d} \qquad \text{w.p. } 1 - 2\exp(-4d)$$

*Proof.* The density of $x$ is defined by the latent Gaussian. Let's denote the error term as $e \sim \mathcal{N}(0, \beta^2 I)$. We have

$$x = G(z) + e$$
$$\implies \|x\| \leq \|G(z)\| + \|e\|$$
$$\implies \|x\| \leq M\|z\| + \|e\|$$

(Where we used assumption 2). Now $z$ and $e$ follow zero-mean Gaussian densities, with variances $I$ and $\beta^2 I$ respectively. Using standard gaussian tail bounds (similar to lemma 21 with $c = 2$), we get the required claim. $\square$

**Remark:**    Since we work under the setting $\beta \leq \mathcal{O}(1)$ from equation 1, we can use $\|x\| \leq \mathcal{O}(\sqrt{d})$ with probability $1 - \exp(\Omega(d))$.

### F.3 CHARACTERIZING LOSS NEAR $\hat{z}$

We need to understand how $L$ and $\nabla L$ behaves near $\hat{z}$ for several claims in the proofs. We will use a Taylor expansion, bounding the contribution of higher order terms.

#### F.3.1 EXPRESSIONS FOR DERIVATIVES

Using equation equation 3, we first calculate the expressions of the derivatives as

$$\nabla L(z) = z + \frac{1}{\beta^2} J_G(z)^T (G(z) - x) \tag{55}$$

$$\nabla^2 L(z) = I + \frac{1}{\beta^2} \left( J_G(z)^T J_G(z) + \sum_{i \in [d]} \nabla^2 G_i(z) \left( G_i(z) - x_i \right) \right) \tag{56}$$

At the inverse point $\hat{z}$, since $G(\hat{z}) = x$, we have:

$$\begin{cases} L(\hat{z}) = \frac{1}{2} \|\hat{z}\|^2 \\ \nabla L(\hat{z}) = \hat{z} \\ \nabla^2 L(\hat{z}) = I + \frac{1}{\beta^2} J_G(\hat{z})^T J_G(\hat{z}) \succeq I \end{cases} \tag{57}$$

Since $\nabla^2 L(\hat{z}) \succeq I$, the function $L(z)$ is strongly convex in the neighbourhood of $\hat{z}$.

#### F.3.2 BOUND ON 3RD DERIVATIVE OF $L$

To show the behaviour of $\nabla L$ around $\hat{z}$, we will first bound the norm of the 3rd order derivatives of $L$ below.

**Lemma 25** (Bounding the 3rd derivative of $L$). *For all $z \in \mathbb{R}^d$, it holds that*

$$\left\| \nabla^3 L(z) \right\|_{op} \leq \frac{\sqrt{d} M}{\beta^2} \cdot \left( 3\sqrt{d} M_2 + M_3 \|z - \hat{z}\| \right)$$

*Proof.* An elementary calculation shows that

$$\nabla^3 L(z) = \frac{1}{\beta^2} \left( 3 \sum_{i \in [d]} \nabla G_i(z) \otimes_T \nabla^2 G_i(z) + \sum_{i \in [d]} \nabla^3 G_i(z) \left( G_i(z) - x_i \right) \right) \tag{58}$$

Here, the tensor product $\otimes_T$ multiplies elements from $\mathbb{R}^d$ and $\mathbb{R}^{d \times d}$ to give elements in $\mathbb{R}^{d \times d \times d}$. We subscript with $T$ to avoid confusion with the kronecker product.

Then, we have:

$$\left\| \nabla^3 L(z) \right\|_{op} \leq \frac{1}{\beta^2} \left( 3 \cdot \left\| \sum_{i \in [d]} \nabla G_i(z) \otimes_T \nabla^2 G_i(z) \right\|_{op} + \left\| \sum_{i \in [d]} \nabla^3 G_i(z) \left( G_i(z) - x_i \right) \right\|_{op} \right) \quad \text{(triangle inequality)}$$

$$\leq \frac{1}{\beta^2} \left( 3 \cdot \sum_{i \in [d]} \left\| \nabla G_i(z) \otimes_T \nabla^2 G_i(z) \right\|_{op} + \sum_{i \in [d]} |G_i(z) - x_i| \cdot \left\| \nabla^3 G_i(z) \right\|_{op} \right)$$

$$\leq \frac{1}{\beta^2} \left( 3 \cdot \left( \sum_{i \in [d]} \left\| \nabla G_i(z) \right\| \right) \cdot \max_{i \in [d]} \left\| \nabla^2 G_i(z) \right\|_{op} + \|G(z) - x\|_1 \cdot \max_{i \in [d]} \left\| \nabla^3 G_i(z) \right\|_{op} \right)$$

$$\leq^{(1)} \frac{1}{\beta^2} \left( 3 \cdot \sqrt{d} \cdot \|J_G(z)\|_F \cdot \left\| \nabla^2 G(z) \right\|_{op} + \sqrt{d} \cdot \|G(z) - x\| \cdot \left\| \nabla^3 G(z) \right\|_{op} \right)$$

$$\leq^{(2)} \frac{1}{\beta^2} \left( 3d \cdot M M_2 + \sqrt{d} \cdot M M_3 \cdot \|z - \hat{z}\| \right)$$

In (1) we use $\sum_{i \in [k]} |\alpha_i| \leq \sqrt{k} \cdot \sqrt{\sum_{i \in [k]} \alpha_i^2}$ on the first part of both the terms; and the fact that $\forall i \in [d]$, $\|T(i)\|_{op} \leq \|T\|_{op}$ for any tensor $T$ (where $T(i)$ denotes a sub-tensor) on the second part of both terms. This fact follows from the sup-definition of $\ell_2$ norm on tensors from (Lim, 2005). In (2), we use Assumption 2, as well as $\|A\|_F \leq \sqrt{\alpha} \cdot \|A\|_{op}$ for $A$ being an $\alpha \times \alpha$ matrix. $\square$

### F.3.3 Perturbation bound of $\nabla L$

Using the above, we show the Taylor expansion result for the gradient of $L$:

**Lemma 26** (Perturbation of $\nabla L$). *For all $z \in \mathbb{R}^d$, it holds that*

$$\nabla L(z) = \hat{z} + \left(I + \frac{1}{\beta^2} J_G(\hat{z})^T J_G(\hat{z})\right)(z - \hat{z}) + R_{\nabla L}(z)$$

*where $\|R_{\nabla L}(z)\| \le \frac{\sqrt{d}M}{2\beta^2} \cdot \left(3\sqrt{d}M_2 + M_3\|z - \hat{z}\|\right) \cdot \|z - \hat{z}\|^2$.*

*Proof.* We Taylor expand $\nabla L$ around $\hat{z}$. Using equation equation 57 we get :

$$\nabla L(z) = \nabla L(\hat{z}) + \nabla^2 L(\hat{z})(z - \hat{z}) + R_{\nabla L}(z)$$
$$= \hat{z} + \left(I + \frac{1}{\beta^2} J_G(\hat{z})^T J_G(\hat{z})\right)(z - \hat{z}) + R_{\nabla L}(z) \tag{59}$$

To bound the *norm* of the remainder term, we note that

$$\|R_{\nabla L}(z)\| = \frac{1}{2!} \cdot \left\|\nabla^3 L(z)\big|_{z=z_{mid}}\right\|_{op} \cdot \|z - \hat{z}\|^2$$

For $z_{mid} = t_{mid}z + (1 - t_{mid})\hat{z}$ for some $t_{mid} \in [0, 1]$. And for the third derivative, we use lemma 25 directly to get

$$\|R_{\nabla L}(z)\| \le \frac{\sqrt{d}M}{2\beta^2} \cdot \left(3\sqrt{d}M_2 + M_3\|z - \hat{z}\|\right) \cdot \|z - \hat{z}\|^2 \tag{60}$$

This finishes the proof. $\qquad\square$

## G Remark on activation functions

The choice of activation functions in our results is largely a matter of convenience. Using standard techniques from approximation theory, e.g. Hornik (1991); Yarotsky (2017), one can approximate a neural network with one choice of nonlinearity via a (comparably sized) neural network with another choice of nonlinearity, under very mild conditions on the nonlinearities. Crucially, this simulation only increases the size by a dimension-independent factor. This result frees us (for purposes of deriving an expressibility result) to work with activation functions chosen for mathematical convenience and produce results that hold without loss of generality.

For instance, if we wished to use ReLU activations, we can use the following lemma. We note that this proof is almost verbatim the same as the proof of Lemma 1.3 in Telgarsky (2017) and proofs for other activations like sigmoid or tanh can be written completely analogously.

**Lemma 27.** *Let $\Omega \subseteq [-M, M]^d$ and let $G_1 : [-M, M]^d \to \mathbb{R}$ be a neural network with at most $l$ layers and $n$ parameters, s.t., the weights $W^{(i)}$ for each layer $i$ and node $j$ in $G_1$ are bounded as, i.e., $\forall i, j : \sum_k |W_{j,k}^{(i)}| \le B$. Furthermore, assume that the activation functions used in $G_1$ belong to the set $\Xi$, s.t. all functions $\sigma : \mathbb{R} \to \mathbb{R}$ satisfy $\sup_{x \in [-B\cdot M, B\cdot M]} \sigma < M$. Then there exists a neural network $G_2$ with ReLU activation and $O(n\frac{(LB)^l lBM}{\epsilon} \log(\frac{(LB)^l lBM}{\epsilon}))$ parameters, such that for all $\sup_{x \in [-M, M]^d} |G_1(x) - G_2(x)| \le \epsilon$.*

*Proof.* For any $\sigma \in \Xi$, from Theorem 1 in Yarotsky (2017) it follows that there exists a neural network $R$ with ReLU activations and $O(\frac{LBM}{\epsilon'} \log(\frac{LBM}{\epsilon'}))$ parameters such that $\sup_{x \in [-B\cdot M, B\cdot M]} |\sigma(x) - R(x)| \le \epsilon'$.

We will construct the network $G_2$ by replacing each activation in $G_1$ with the corresponding network $R$ as given by the result above with $\epsilon' = \epsilon/l$. Note, this network is at most a factor of $O(\frac{(LB)^l lBM}{\epsilon} \log(\frac{(LB)^l lBM}{\epsilon}))$ bigger than $G_1$, as the lemma requires.

We will prove the claim of the lemma by induction on $l$. More precisely, we will show (by induction) that for each node at layer $i$, the network $G_2$ calculates a function that is $(LB)^i i\epsilon'$ away in $l_\infty$ norm from the corresponding node in $G_1$, and the inputs to the node are in $[-BM, BM]$.

For the base case $i = 1$, since the input $x \in [-M, M]^d$, the result follows by Theorem 1 in [Yarotsky (2017)].

We proceed to the inductive claim. Let $H(x)$ denote the vector valued mapping computed by the nodes at layer $i$, and let $H_R(x)$ be the corresponding vector in $G_2$. As inductive hypothesis, we assume that $\|H(x) - H_R(x)\|_\infty \leq (LB)^i i \epsilon'$ for all $x \in [-M, M]^d$ and $\|H(x)\|_\infty \leq M$ as well as $\|H_R(x)\|_\infty \leq M$. Therefore, for the $j^{th}$ node in layer $(i+1)$ in network $G_1$ we have $|W_j^T H(x)| \leq \|W_j\|_1 \|H(x)\|_\infty \leq BM$ and $\sigma$ is bounded by $M$ on this interval, so we have $\|\sigma_1(W_j^T H(x))\|_\infty \leq M$. Along with the bound on the activations, the part of the inductive hypothesis about the size of the input is proven. To prove the error bound, we have:

$$|\sigma_1(W_j^T H(x)) - R(W_j^T H_R(x))| \leq |\sigma_1(W_j^T H(x)) - \sigma_1(W_j^T H_R(x))| + |\sigma_1(W_j^T H_R(x)) - R(W_j^T H_R(x))|$$
$$\leq L|W_j^T(H(x) - H_R(x))| + \epsilon'$$
$$\leq L\|W_j\|_1 \|H(x) - H_R(x)\|_\infty + \epsilon'$$
$$\leq (LB)^{i+1}(i+1)\epsilon'$$

This finishes the proof of the inductive step, and thus the lemma. $\qquad\square$

