# OpenReview forum: "The Effects of Invertibility on the Representational Complexity of Encoders in Variational Autoencoders "
_ICLR.cc/2022/Conference — ICLR 2022 Poster_

### Official Review · Reviewer_agVY · 2021-10-24

**Correctness:** 4
**Technical Novelty And Significance:** 2
**Empirical Novelty And Significance:** 3
**Recommendation:** 6
**Confidence:** 3

**Main Review:**

Unfortunately, the exploration made appears to me mostly about something else. This whole paper doesn’t read like a paper about VAEs, or about DL, but rather about applying a conjecture by Katz & Lindell (2020), having to do with Boolean circuits, to the situation of VAEs.

In that regard, the situation studied about VAEs feels rather artificial because in practice we never deal with random mappings or with worst-case distributions. The whole edifice of deep learning is predicated upon assumption that patterns in the experimental data will play nicely with the inductive bias of neural networks. This does not invalidate the study of worst-case scenarios, but it makes it less interesting.

In fact, the whole paper rests upon the aforementioned conjecture coming from the world of cryptography, for which not much background is given. For all the reader can tell, that conjecture cited with reference to a book that came out in 2020, could be false. That would make this very paper completely void.

After having said all that, I’ll admit to my lack of familiarity with some of the techniques used by the authors despite being knowledgeable about VAEs.

**Summary Of The Paper:**

The authors of the paper start from a great initial question that connects the complexity of the encoder and generator of a VAE. It is reasonable to suspect that the complexity of one mapping would have an effect on the complexity of the other mapping.

**Summary Of The Review:**

The paper starts with a good inquiry, but the answer provided is unsatisfying. It relies on a conjecture from another field which, even if proven true, would not make it all that useful.

---

> ### Author Response · Authors · 2021-11-10
> **Thank you for the helpful feedback!**
>
> We thank you for your helpful feedback!
>
> It seems like you have two main sources of concern, which we hope to alleviate:
>
> (1) The citation for the one-way function conjecture being recent (2020), and how plausible it is.
>
> (2) How much of the paper relies on and is about the one-way function conjecture, and the relevance of the conjecture to “real-life” trained VAEs.
>
> Regarding (1): The existence of one-way functions is widely believed in the cryptography community, and the formal statement of the conjecture can be traced to the famous 1976 paper on the Diffie-Hellman public key cryptosystem [1]. The reference we provided (Katz & Lindell) is a classic textbook, the most recent edition of it being 2020. We think it’s not an overstatement to say that the hypothesis has a similar status of believability among experts as $P \neq NP$.
>
> Regarding (2):
> Note that our paper has *two* main results: Theorem 1 (in brief) shows that when the mean map of the generator is invertible, we can construct an approximate encoder of a comparable size as the generator; Theorem 2 (in brief) shows that if we drop invertibility, *there exist* generators—whose construction depends on the existence of one-way functions—for which the encoders need to be exponentially bigger. Theorem 1 does not have anything to do with one-way functions, and is a constructive result (we construct a good encoder given the generator, s.t. the size of the encoder is comparable to the generator). Moreover, this construction elucidates connections between invertible generators, Langevin diffusion and hierarchical encoders, and is likely to be the more interesting result for generative model practitioners.
>
> The message of Theorem 2 is that if the generator is *not* invertible, some *additional* structure needs to be imposed if we wish to have tractably sized encoders. Thus, the purpose of the construction of Theorem 2 is not to match "real life" VAEs but to show non-invertible generators—unless otherwise restricted—can require much larger encoders. We completely agree with you that the result of Theorem 2 does not preclude some other structure in the data making the problem tractable.
>
> Another way to say this is that the takeaway of Theorems 1+2, taken together, is that invertibility of the generator mean map is one assumption that can lead to tractably sized encoders. We certainly do not claim it’s the *only* one (which is the point that you are raising).
> We do believe that it’s valuable for theoretical research to address *what kind* of structure in the data makes it so that VAEs (or other generative models for that matter) can efficiently easily learn it. We also hope that you’ll agree that identifying potential sources of tractability generally proceeds one paper at a time—i.e. we cannot identify all possible structures in data that could make it amenable to being learnt by VAEs or other deep generative models.
>
> [1] New directions in cryptography. Diffie, Whitfield, and Hellman, Martin.  IEEE transactions on Information Theory 22.6 (1976): 644-654.

---

> > ### Comment · Reviewer_agVY · 2021-11-15
> > **thanks for the response**
> >
> > I'll adjust my review slightly, based on the authors' response. I like the analysis that they have done, and especially the novelty they brought. I think it's pretty much unusable in practice (sadly), as another reviewer pointed out, but I'd like to see more papers that push in that kind of direction (and especially import results from adjacent fields). I'm not sure how this theory could be presented in order to be more accessible to other deep learning researchers.
> >
> > I have noticed three typos while re-reading the latest version of the paper:
> > - "isn’t" in the abstract, please don't use contractions in a scientific paper,
> > - "when mean of the generative" should be "when the mean",
> > - "G upto order 3"

---

> > > ### Author Response · Authors · 2021-11-17
> > > **Thank you for reconsidering your score!**
> > >
> > > Thank you for your encouraging words, as well as reconsidering your score! If you think of any, we would definitely appreciate suggestions to make the paper more accessible.
> > >
> > > There is a sense in which we agree that the paper isn't *immediately* useful for practitioners: we don't provide any immediate advice to practitioners on how to improve training of VAEs. We do think though, given that variational methods are predicated upon having tractably sized encoders, and given that a substantial part of the VAE community believes that complicated posteriors is an obstacle to VAE training—it is important to identify conditions under which tractably sized encoders exist. The condition we identify is that the data can be written as the output of a small generator with an invertible mean map. Again, there is a sense in which this is not surprising given practitioner wisdom/experience: it is a common belief that data which is "robustly full dimensional" (which is in some sense what an invertible generator results in) is easier than data which lies on a complex enough low dimensional manifold. Our results formalize some part of this intuition, and the proof of this fact is in our opinion far from obvious.
> > >
> > > We agree though that we'd like to also see a lot more theoretical work in quantifying what kind of data is "easy" and what kind of data is "hard" for different generative model families, not just VAEs.

---

### Official Review · Reviewer_8CZc · 2021-10-27

**Correctness:** 3
**Technical Novelty And Significance:** 2
**Empirical Novelty And Significance:** 2
**Recommendation:** 6
**Confidence:** 3

**Main Review:**

Pros:
- The idea of using Boolean permutations as a proof method in network analysis context in the presented manner appears novel to me.
- The said idea might well be useful to prove other network properties.
- The mathematics, to the extent that I could follow it, seems correct.
- The paper does seem to prove the claimed complexity properties of inference models in the "approximately invertible" and the "non-invertible" cases.
- The paper puts together an impressive amount of math to prove the points, with extensive details in the supplement (the latter of which, however, I did not have time to evaluate in detail)

Cons:
- The paper suffers from confusing structure and sloppy language at critical points. It starts with sweeping questions such as 'How to choose the architecture' and 'When is inference (much) harder than generation', but then actually focuses only on very specific and narrow parts of those wide questions. The wide questions are then followed by a moderately heavy load of math. The paper reads as a string of theorems and remarks, where the reader has to jump around to understand how the theorems relate to the overall points of the paper. The proof is somewhat complicated and it is difficult to confirm the derivation holds at every step (though I admit this final point may be a failure from my side).
- VAE basics, for example, are given considerable amount of space, but there is no ending summary or Conclusion to draw things together, which adds to the impression that the ultimate significance of the work is not clear.
- There is no empirical part nor clear indication of empirical usefulness of the results. Both theorem 1 and theorem 2 prove the *existence* of networks that satisfy the conditions. Could the authors help me to understand the connection to empirical significance, or should we simply approach this as a mathematical exercise?
- The final sweeping claim in the introduction, namely that the paper lends theoretical support to the idea that "learning deep generative models more generally is harder when data lies in a low-dimensional manifold" just seems inappropriate to me. I'm not sure the whole notion of 'learning more generally when data lies in a low-dimensional manifold' makes sense as an expression. Being familiar with the cited (Arjovsky, 2017), I believe I understand what the authors mean, but I don't see how their results really lend the said support to that idea. Could the authors explain this, or point to where is this explained?
- In terms of impact, first, it is not surprising that the complexity requirements of inference and generative subnetworks are similar in the typical case, and in fact, the authors acknowledge this, but present the value in this work for providing theoretical foundation for this "empirical wisdom". I find this work somewhat valuable in this regard, but I did not quite understand, do the theorems actually give us a way of telling, for a given network, whether it has the strongly invertible nature or not?
- Some of the citations are not entirely appropriate. E.g. in the first paragraph, the ideas cited for both (Betherlot,2018) and (Shen,2020) have certainly been presented before those papers came out; e.g., the modification of latent space has been investigated at least since [1] in a similar context.
- The very place in the text that could illuminate the significance of the results (Sec. 1) reads as "We identify...an aspect...: a notion of approximate bijectivity of the mean of the generative direction. We prove that under this assumption..." Forgive me, but what does it mean that one 'identifies an aspect' which turns into 'notion', and then 'proves [things] under this assumption'? This is just unnecessarily fuzzy, in striking contrast with the mathematical precision shown in some other parts of the paper.
- The results of Lipton & Tripathi (2017) do not show that "these architectures are invertible in practice" as the authors state. What they may have shown is that those architectures are *sometimes* invertible in practise. E.g. in GAN context, AFAIK the general inversion of CNNs is an open research problem. If I am wrong, I ask the authors to explain why.

[1] Salimans et al. 2016, Improved Techniques for Training GANs.



**Summary Of The Paper:**

This purely mathematical paper investigates the important question of how does the necessary level of complexity of an inference subnetwork depend on the the complexity of the corresponding generative subnetwork, in a VAE model. The paper introduces a specific measure of invertibility, and uses it to show that a generative subnetwork that is sufficiently strongly invertible only requires an inference model at similar complexity, while some "non-invertible" generative subnetworks will require an exponentially more complex inference subnetwork. The paper provides a lengthy and comprehensive proof of both.

**Summary Of The Review:**

The proofs in the paper appear correct, and at first glance, they appear relevant to generative models research community.
However, upon closer look, despite the commendable theoretical work, the paper seems to amount to proving the *existence* of a network of certain complexity, in the invertible case, and the *existence* of a generative subnetwork that only allows inference networks with pathological (exponentially growing) complexity. Without any guidance of how one could construct the networks under the said conditions, or measure how far we are from meeting those conditions, I am worried that the paper boils down into a mathematical exercise without considerable impact in this line of research.

I also had concerns about the sloppy wording of the claims in the introductory section, and the structure of the paper in terms of the order of how things are presented, and the lack of clear bottom line.

In combination, I lean towards rejection, but I may improve my score if the authors can convince me of the significance of the theorems, and if they manage a considerable rewrite of the introduction section. Given the considerable theoretical weight and intellectual import of the paper, I would encourage the authors to do so.

---

> ### Author Response · Authors · 2021-11-12
> **Thank you for the detailed feedback! 1/2**
>
> We thank the reviewer for the very detailed and thoughtful review! We hope to address your concerns regarding the exposition and are confident this will make the final version of the paper more impactful! We have also revised and updated several parts of the submission according to your comments—we summarize the changes next to the responses below.
>
> **Regarding the main takeaway and significance of the results**:  Our goal is to initiate a study of the complexity of inference (i.e. the size of a model approximating the posterior distribution in a latent-variable model) as a function of the complexity of the generative direction of the model. The reason this question is relevant is that in variational approaches to training latent variable models, the posterior needs to be approximable by a reasonably sized model, in order to be able to efficiently train a reasonably tight ELBO bound.
>
> We focus on VAEs as a class of generative models, where there is a well-defined notion of complexity (i.e. the size of the network representing the generator and encoder). We then identify invertibility of the mean map of the generator as a property that ensures the existence of a compact encoder (i.e. of size comparable to the generator). This is the content of Theorem 1. Conversely, if the mean map is not invertible, there exist some generators for which a good encoder (i.e. approximating the posterior accurately) needs to be exponentially larger. This is the content of Theorem 2.
>
> *The takeaway of Theorems 1+2, taken together*, is that if the data we are training our model on is such that it can be written as the output of a small generator with an invertible mean map, we can hope that training with a (comparably small) encoder can be successful. Methodologically, the goal of this kind of theoretical approach is to explore what kind of structure in the data makes it so that VAEs (or other generative models for that matter) can efficiently learn it.
>
> At the level of conceptual insights, the construction of the encoder in Theorem 1 reveals a connection between diffusion processes (more precisely Langevin diffusion) and hierarchical encoders—the construction of the encoder essentially simulates a step of Langevin diffusion in each of the layers of encoder.
>
> We have edited the introduction to make this high level motivation, as well as takeaways more clear. We have also added a short conclusion section where we summarize these points and suggest some future directions.
>
> **Regarding informal language in the introduction and technical nature of the rest of the paper**: Our goal was to make the introduction read as exposition (thus, the language is loose and not completely formal) and motivate the high-level question we are asking—namely, when is the complexity (i.e. # of parameters) of a network performing inference in a latent-variable model (i.e. an encoder approximating the posterior) comparable to the complexity of the generator. To the best of our knowledge, this question has not been asked before for any model (VAE or otherwise), so we dedicated some space motivating the high-level motivation behind it: having a low-complexity encoder is crucial to have a efficiently tractable and tight ELBO bound in any latent-variable model trained using a variational approximation.
>
> On the other hand, even formalizing our claims, let alone sketching the proofs requires a certain amount of technical definitions and lemmas, which is why the main part of the paper reads quite technical.
>
> We understand though how some of the exposition might have come off as overly ambitious and grandiose, so we tempered several crucial sentences in the updated draft.
>
> **Regarding the existence of encoders vs training them**: You are right, our paper does not touch on questions of training dynamics (i.e. what kinds of generators/encoders gradient descent would converge to). This is not unusual: typically questions of expressivity (i.e. how large of a neural network is needed to approximate a function/distribution with certain properties) are much easier to answer than questions of training dynamics (which involves understanding a highly non-convex optimization process). Even in standard supervised learning, understanding what kinds of optima gradient descent converges to is very poorly understood, but the expressive power of neural nets with various restrictions (depth, activation functions, etc.) is much better understood. To the best of our knowledge, our paper is the *first* to explore the (representational) complexity of the encoder as a function of the complexity of the generator. We certainly hope more work will follow on understanding questions of training dynamics.

---

> ### Author Response · Authors · 2021-11-12
> **Thank you for the detailed feedback! 2/2**
>
> **Regarding hardness of learning data supported on low dimensional manifolds**: When the mean map is not invertible, the distribution the generator produces does not have full support (i.e. is supported on a lower dimensional manifold). Thus, Theorem 2 shows that there are *some* distributions supported on low dimensional manifolds, for which a good encoder will be very large. We have tempered and clarified the language in both the abstract and introduction of the paper.
>
> **Regarding Lipton-Tripathy (2019)**: in the paper, the authors show that simple gradient descent with gradient clipping recovers a preimage 100% of the time. It’s true that this is the case on a GAN trained on real data—there might be *some* networks for which finding a pre-image is hard. (In fact, Lei et al (2019) show this is NP-hard in general. Though it seems GAN training does not recover such "worst case" networks.)
>
> **Regarding attributions of citations**: thank you! We have added your suggested citation.
>
> Qi Lei, Ajil Jalal, Inderjit S Dhillon, and Alexandros G Dimakis. Inverting deep generative models,one layer at a time. NeurIPS 2019

---

> ### Author Response · Authors · 2021-11-19
> **Have we addressed your concerns?**
>
> We thank the reviewer again for the detailed review. As the review period is slowly drawing to a close, we wish to followup with the reviewer regarding both the writing and significance concerns. In particular, our updated draft substantially reworked the introduction according to the reviewer's suggestions and questions.

---

> > ### Comment · Reviewer_8CZc · 2021-11-22
> > **Thank you for the responses**
> >
> > I thank the authors for their clarifications. I largely consider my concerns addressed. I remain sceptical about the ultimate significance of the results, but as I stated in the beginning, I like the general approach taken in proving the theorems.
> >
> > Given the technical ambition and other merits of the paper, and the improvements made to the manuscript, I have increased my score towards acceptance.
> >
> > A final minor note: I still believe that the authors misrepresent the results of Lipton & Tripathi. I suggest that the wording is yet revised to sound less sweeping. The phrase "these architectures are invertible in practise" seems to imply that the inversion problem is solved "in practise", while it most definitely is not.

---

> > > ### Author Response · Authors · 2021-11-22
> > > **Re: Thank you for the responses**
> > >
> > > Thank you for your response, encouraging words, and reconsidering your score! We will tone down the wording regarding Lipton-Tripathi a bit more.

---

### Official Review · Reviewer_hddw · 2021-11-01

**Correctness:** 4
**Technical Novelty And Significance:** 4
**Empirical Novelty And Significance:** Not applicable
**Recommendation:** 6
**Confidence:** 3

**Main Review:**

This is a purely theoretical paper trying to solve a very interesting and important question, which is generally a good paper.  However, I still have some concerns and questions about it.

Pros:
1. This paper is well-written and intuitive. It's difficult to achieve that for a purely theoretical paper.
2. The proofs are reasonable. Sorry I only checked the main theorem and some lemmas and didn't find the time to delve into the proofs in the appendix.

Concerns and Questions:
1. The inference model is assumed to be a deep latent Gaussian model, which is not the case for VAEs. The structure looks similar to some hierarchical VAE but it's not the same. This heavily limits the usage of the conclusion in usual VAEs.
2. In remark 3, the authors mention that the case for layerwise invertible models is very simple. They argue that some convolutional NN contains operations that increase and then decrease the dimensions, which makes layers not invertible and necessitates the invertible rather than the layerwise invertible assumption. But why not regard the two layers as a single layer, which becomes invertible again? I guess it is not difficult to get a reverse of such a layer.
3. The paper mentions that Learning deep generative models more generally is harder for data on low-dimensional manifolds. This paper (https://arxiv.org/abs/2106.13746) gives some explanation why deep generations models can deal with some manifolds easily but fail on others. Can we use the method in your paper to analyze the problem as well?

**Summary Of The Paper:**

The paper answers the question how complex inference models need to be to accurately estimate posterior distributions. The conclusion is when a latent Gaussian model with N parameters satisfies (1) strong invertibility, (2) 3-th smoothness, the posterior can be approximated by a deep latent Gaussian model with O(N) parameters. A special case is also given to show that strong invertibility is necessary to the conclusion.

**Summary Of The Review:**

The paper is well-written and the conclusions are reasonable. But I still have some concerns about the too strong assumptions of the main theorem, which might limit the usage of the conclusion.

---

> ### Author Response · Authors · 2021-11-10
> **Thank you for the thoughtful review!**
>
> We are very glad you liked our paper, both in terms of content and quality of writing, and thank you for the thoughtful feedback!
>
> To address your questions/concerns:
> 1. Modern architectures indeed frequently feature hierarchical encoders (e.g. [1]). Note that an encoder with a single stochastic layer necessarily produces a *Gaussian* posterior—so it necessarily will be a bad approximation to the true posterior when it’s far from Gaussian. It is certainly an interesting question to explore when this *is not* an issue for training a good decoder (i.e. the encoder does not provide a good ELBO bound, but it is still good enough to learn a good decoder)—but this is not a question we tackle in this paper.
> 2. This is a good question, and we will clarify in the paper: the “simple” algorithm in Remark 3 for inverting a composition $f_1 \circ f_2 \dots \circ f_n$ of invertible layers consists of simply applying $f_n^{-1} \circ f_{n-1}^{-1} \dots \circ f^{-1}_1$.  If, however, $f_i \circ f_j$ as a whole happens to be invertible (e.g. $f_i$ maps to a higher dimensional output, then $f_j$ maps to a lower dimensional output, but is invertible on the image of $f_i$), without each of $f_i,f_j$ individually being invertible, it is not clear how to write down an inverse explicitly in terms of $f_i, f_j$. Of course, one could use our algorithm (doing gradient descent on the preimage) and proving this works is in fact the content of Lemma 1.
> 3. There have indeed been many variants of VAEs intended to deal with data with low-dimensional support, by either changing the priors as in the paper you linked (see additionally [2]) or changing the learning algorithm (e.g. the two-stage process in [3]). To our knowledge none of the algorithms come with any formal guarantees. We agree that understanding *what* manifolds are easier to learn from is a very interesting question.
>
> [1]: NVAE: A deep hierarchical variational autoencoder, Vahdat, Arash and Kautz, Jan. NeurIPS 2020.
>
> [2]: Variational Autoencoders with Riemannian Brownian Motion Priors. Kalatzis, Dimitrios and Eklund, David and Arvanitidis, Georgios and Hauberg, Soren. ICML 2020.
>
> [3]:  Diagnosing and Enhancing VAE Models. Dai, Bin and Wipf, David. ICLR 2018.

---

> > ### Comment · Reviewer_hddw · 2021-11-20
> > **Thanks for the responses**
> >
> > I thank the authors for answering all my questions. But I still think the analysis of the paper~(especially the main theorem) can only be applied to very limited kinds of VAEs.
> > For example, the authors mentioned NVAE in their response. However, theorem 1 assumes a latent Gaussian decoder, which is not applicable to NVAE, which has a hierarchical decoder. Let alone that NVAE shares the latent structures between its encoder and decoders.

---

> > > ### Author Response · Authors · 2021-11-20
> > > **Re: Thanks for the responses**
> > >
> > > Thank you as well!
> > >
> > > We would just like to note that the reason we mentioned NVAEs in point 1 in our rebuttal was to point them out as an example of an architecture that uses a hierarchical encoder. Our paper isn't intended to specifically address the NVAE architecture.
> > >
> > > That being said, the encoder that is constructed in Theorem 1 *is* in fact implicitly weight tied to the generator, as the stochastic maps in the encoder each have mean function $z \to z - \eta J_G(z)^T(G(z) - x)$ where $G$ is the generator mean map and $J_G$ is the Jacobian of $G$. (Though this is not the same weight-tieing schema as in NVAEs.)
> > >
> > > Furthermore, it is quite conceivable that our proof techniques can handle a hierarchical generator (i.e. a deep latent Gaussian) so long as each of the stochastic layers have an invertible mean map and a sufficiently small variance (i.e. $\beta_i$ is sufficiently small)—we just didn't consider this extension crucial, especially for a first analysis of the kind we provide.

---

> > > > ### Comment · Reviewer_hddw · 2021-11-20
> > > > **Re:Re: Thanks for the responses**
> > > >
> > > > Thanks for the clarification! I would recommend the authors to include several concrete examples where the theory applies in their next version. Also, some empirical studies might also be beneficial to verify the proposed theory on real problems.

---

### Decision · Program_Chairs · 2022-01-20

**Decision:**

Accept (Poster)

**Comment:**

All three reviewers viewed this paper as marginally above the acceptance threshold (6).

Most of the initial concerns of reviewers were around (a) the applicability of the theory to actual practical use cases and networks, and (b) the presentation and framing of the work, and scope of its results. There were fairly detailed responses from the authors: two of the three reviewers increased their scores after the author response. There's still some lingering questions as to how "real-world" relevant the theory is, but the consensus at this point is to accept the paper.

My primary concern for acceptance would be that the proofs techniques are based on Boolean circuits, and none of the reviewers (nor the AC) are particularly familiar with this, and thus the proofs in the appendix have been only lightly reviewed. The "impression" of all reviewers is of correctness.